



**1** **Influence of intra-event rainfall variation on surface–subsurface flow**

**2** **generation and soil loss under different surface covers by long-term**

**3** **field observations**

Jian Duan[a*], Hai-Jin Zheng[a], Yao-Jun Liu[b], Ming-Hao Mo[a], Yue-Jun Song[a], Jie Yang[a]
[a] *Jiangxi Provincial Key Laboratory of Soil Erosion and Prevention, Jiangxi Academy of Water*
*Science and Engineering, Nanchang, Jiangxi 330029, China*
[b] *School of Geographic Sciences, Hunan Normal University, Changsha, 410081, China*
**\* Corresponding author:**
Jian Duan
Institute of Soil and Water Conservation,
Jiangxi Academy of Water Science and Engineering,
Nanchang, Jiangxi 330029, China.
E-mail: djlynn20@126.com
Tel: + +86-0791-87606870
Fax: +86-0791-87606699



**Abstract:** Rainfall is the main driver of runoff generation and soil erosion. The impacts of natural
rainfall on water erosion have been extensively studied at an inter-event scale; however, very few
studies have explored the intra-event influences and associated responses to different surface
cover types. In this study, long-term in situ field observations of surface runoff, subsurface flow,
and soil loss characteristics in three surface cover plots (bare land, litter and grass cover) under
natural rainfall events were conducted from 2002 to 2012 in the red soil hilly region of southern
China. According to the period of most concentrated rainfall, 262 rainfall events were classified
into four types of intra-event variation: advanced, intermediate, delayed, and uniform patterns. For
bare land, the advanced pattern with the shortest duration and the highest intensity was main
rainfall type for surface runoff and soil loss; the contribution rates were 57.24% and 75.17% for
surface runoff and soil loss, respectively. Sediment yields were more sensitive to intra-event
rainfall variation than surface runoff. The highest subsurface flow was found in the delayed
pattern with the longest duration and high depth, followed by the uniform, intermediate, and
advanced patterns. For all rainfall patterns, compared to the bare land, surface cover significantly
reduced surface runoff and soil erosion by 88.01 to 91.69% and by 97.80 to 97.95%, respectively,
while subsurface flow was increased from 3.55 to 5.92 times. The reduction benefits of litter cover
were comparable to those of grass cover. However, the increasing benefit of subsurface flow for
litter cover for each rainfall pattern ranged from 1.38 to 2.67 times those of grass cover. Moreover,
surface cover weakened the influences of intra-event rainfall variation on surface-subsurface flow
and soil loss. The results demonstrated that intra-event rainfall variation had important effects on
surface-subsurface flow and soil loss, and provided a basis for optimizing surface cover measures
to effectively respond to extreme water erosion and drought caused by global climate change.





**Keywords:** subsurface flow; rainfall intensity fluctuation; surface runoff; soil erosion; natural

rainfall; surface cover

**1 Introduction**

Soil is a crucial natural resource for life on Earth, similar to water and air. Soil offers a wide range

of services and products, especially ones on those that support biodiversity, water cycling, soil

conservation, carbon sequestration, and ecosystem productivity. Soil degradation is becoming a

highly serious risk to land productivity and human well-being (Pimentel et al., 1976; Montanarella

et al., 2016). A major cause of soil degradation is water erosion, resulting in losses in topsoil and

nutrients (Poesen, 2018; Tsymbarovich et al., 2020). Numerous investigations have noted a

decline in soil quality in various regions throughout the world. This decline helps to explain why

production costs are increasing, crop yields are declining, and farmland is even being abandoned

in worst-case scenarios (Montgomery, 2007; García-Ruiz et al., 2015). According to the Food and

Agriculture Organization (FAO)-led Global Soil Partnership, 75 billion tons of soil is lost from

agricultural lands globally each year, causing an estimated $400 billion in annual economic losses

(GSP, 2017).

Rainfall is the main driver of runoff generation and soil erosion. Inter-event rainfall variables,

such as total rainfall amount (RA), rainfall duration (RD), average intensity (I), and maximum

30-min intensity (I30) are frequently utilized to evaluate the impacts of inter-event rainfall

characteristic variability on runoff and soil erosion (Hammad et al., 2006). Some researches

employ rainfall amount, rainfall duration, and rainfall intensity (I30) as indicators to translate

rainfall events into distinct rainfall regimes, such as long-duration/light-intensity and

short-duration/heavy-intensity regimes, to explore water erosion characteristics (Wei et al., 2007;



Fang et al., 2012). These rainfall parameters, however, do not account for intra-event variability
characteristics in natural rainfall features, such as the temporal distributions of intensity peaks in
rainfall profiles (Flanagan et al., 1988), which have important impacts on water erosion and
related landsurface processes (Dunkerley, 2012; An et al., 2022; Liu et al., 2022). For example,
rainfall events with the same features (e.g., RA, RD, I, and I30) create significantly different
runoff rates and soil losses, and these phenomena may be caused by intra-fluctuations in rainfall
characteristics (Todisco, 2014; Mohamadi and Kavian, 2015; Almeida et al., 2021). Therefore,
there is a strong need to incorporate intra-event rainfall variability when studying rainfall
infiltration, runoff generation, and soil erosion (Dunkerley, 2012; Dunkerley, 2021b).
Numerous studies have shown that intra-event rainfall variability have significantly impacts on the
processes of soil erosion, particularly on runoff, soil loss, and particle dispersion (Zhang et al.,
1997; Parsons and Stone, 2006; An et al., 2014; Wang et al., 2017). The results of previous studies
have largely been obtained based on simulated rainfall experiments. For example, the rainfall
duration is generalized into three equal periods, and combinations of different rainfall intensities
are designed by tuning several intensity peaks to simulate rainfall intensity patterns, such as
increasing, decreasing, rising-falling, falling-rising and constant patterns (Wang et al., 2017;
Alavinia et al., 2019; Macedo et al., 2021). However, the simplified synthesis approach may not
adequately capture the complexities of intra-fluctuation in natural rainfall (Wang et al., 2016;
Macedo et al., 2021; Liu et al., 2022). Natural rainfall events are notable for their continually
variable intensities, making it difficult to extend previous research based on simulated rainfall
conditions to natural rainfall conditions. Due to the lack of high temporal resolution data in natural
rainfall from long-term observations, previous studies cannot accurately reveal the runoff and



erosion differences caused by intra-event rainfall variability. Furthermore, most of the existing
research focuses on the impacts of inter- and intra-event rainfall differences on surface runoff and
soil erosion (Parsons and Stone, 2006; Wang et al., 2017; Alavinia et al., 2019), and little attention
is given to the response of subsurface flow generation. Subsurface flow is a key component of
rainfall runoff, and its output is even higher than that of surface runoff in the rainfall regime with
long duration and high depth (Liu et al., 2016; Duan et al., 2017). The formation of subsurface
flow altered soil moisture redistribution, soil hydrology and slope erosion processes (Zheng et al.,
2004; An et al., 2021). The knowledge gaps impede a better understanding of soil hydrological
processes and erosion mechanisms caused by natural rainfall.
To reduce water erosion, mulching with litter or living plants is widely used around the world to
increase surface coverage (Shi et al., 2012; Duan et al., 2022). Surface cover with litter or living
plants effectively reduces the kinetic energy of raindrops and protects the soil surface from
raindrop splashing. On the other hand, surface cover roughens the surface and causes overland
flow tortuosity, which boosts infiltration and lessens water erosion (Nearing et al., 2005). The
essential roles of surface cover in promoting rainfall infiltration and reducing surface runoff and
soil loss have received widespread attention and positive evaluation, but little attention has been
paid to the role of surface cover in regulating surface-subsurface flow and soil loss under different
intra-event rainfall variations. As a result, a scientific assessment of the effects of surface cover on
surface-subsurface flow and soil erosion under intra-event rainfall variations are critical for
watershed flood prediction and forecasting, soil erosion, and hydrological cycle computation.
By considering the above knowledge gaps, in this study, long-term in situ field observations of
surface-subsurface flow and soil loss were conducted in the red soil hilly region of southern China



from 2002 to 2012 for three surface cover types (bare land, litter and grass cover) under natural
rainfall conditions. Based on rainfall data with a 1-min temporal resolution, 262 rainfall events
over 11 consecutive years were selected and classified into four types of intra-event rainfall
variation including advanced, intermediate, delayed, and uniform. The purpose of this study was
to (1) investigate whether intra-event rainfall variability influences surface-subsurface flow and
soil loss under natural rainfall conditions; (2) explore the effects of different surface cover types
on surface-subsurface flow and soil loss, and (3) determine the role of surface cover in regulating
surface-subsurface flow generation and soil loss for intra-event rainfall variation.
**2 Materials and Methods**
*2.1 Study area*
The red soil hilly regions are located in the tropical and subtropical regions of China, with a total
area of 1.18 million km$^2$. This region is an important agricultural area for tropical and subtropical
fruits, cash and food crops in China. However, the region has experienced severe water erosion
due to intense rainstorms, the hilly terrain and unsustainable human activities (Duan et al., 2022).
Many vegetation restoration projects have been implemented since the 1980s. Surface cover with
living grass and litter is an effective technique to prevent water erosion, and it is widely used for
soil and water conservation efforts. The eroded area and degree were effectively controlled in this
region.
The study was conducted in the Jiangxi Eco-Science Park of Soil and Water Conservation, which
is located in the Yangou catchment (29°16′ N to 29°17′ N, 115°42′ E to 115°43′ E), 15 km away
from the largest freshwater lake (Poyang Lake) in Jiangxi Province, southern China (Fig. 1). The
catchment has a subtropical humid monsoon climate with an average annual precipitation of 1469




mm. The rainfall distribution is uneven throughout the year, with approximately 70% or more of
the total precipitation falling between spring and summer (from March to August). The altitude of
this area ranges from 30 m to 90 m, and the mean annual temperature is approximately 16.7 °C.

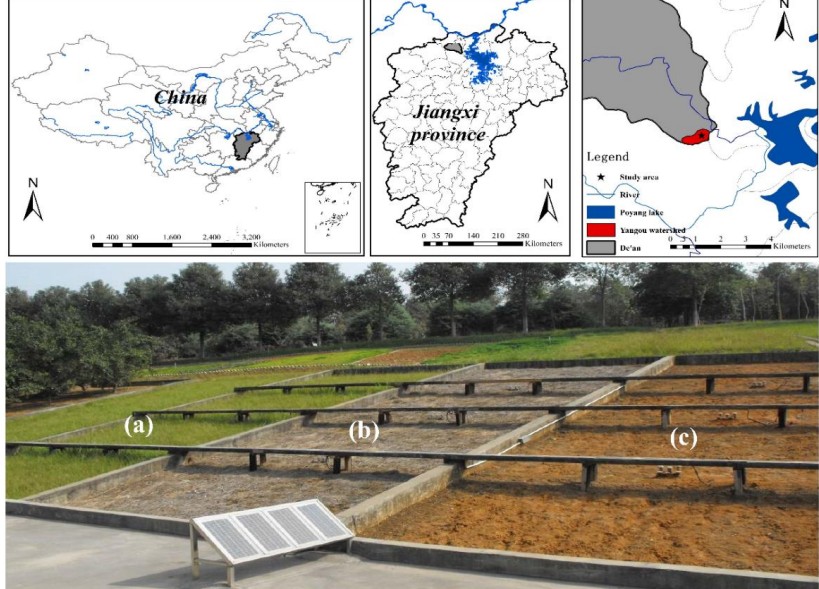


**Fig. 1** Location of the study area and the photos of three types of runoff plots including grass
cover (a), litter cover (b) and bare land (c).
The dominant soil type of the region is red clay soil, which is formed by the decomposition of
Quaternary sediments. Red clay soil is classified as Ultisol in the USDA soil taxonomy system,
and it is vulnerable to water erosion. This soil has a texture composed of 11.54±1.21% sand
(2-0.05 mm), 68.06±0.15% silt (0.05-0.002 mm), and 20.41±1.19% clay (<0.002 mm). The soil
thickness typically exceeds 100 cm, and the soil profile type is Ah-Bs-Cs according to Soil
Taxonomy (Liu et al., 2016; Ma et al., 2022a). The soil physicochemical properties vary
considerably among the different layers, especially regarding soil porosity and water infiltration
capacity. The topsoil layer (Ah) is typically 30 cm, and it is susceptible to severe soil erosion
because of its loose structure (1.27±0.10 g cm$^{-3}$) and the high precipitation in this region. The


depth of the Bs layer is 30-60 cm with a compact structure (1.42±0.08 g cm$^{-3}$) and low
permeability. The soil below 60 cm is defined as the Cs layer (parent material) with a tight
structure (1.53±0.07 g cm$^{-3}$) and poor permeability.
*2.2 Plot construction*
Three in-situ runoff plots with a size of 15 m × 5 m (length × width) were built on an open
south-facing hillslope. Since soil erosion originates mainly from steep slopes in the red soil hilly
regions, the runoff plots were set to a fixed slope of 14° based on field observations. The adjacent
plots were isolated by 100 cm high concrete walls to prevent hydrological interference. Different
soil layers are responsive to hydrological processes, such as surface and subsurface flow under
natural rainfall events. To accurately measure the subsurface flow, a L-type steel plates with a gap
of 5 cm gaps were set up at a depth of 60 cm (Fig. 2). Porous nylon gauze was used to separate the
soil from the steel plates. A concrete wall was constructed outside the steel plates and a certain
area was left below the plates as a trench, which was connected to a container through a plastic
pipe to collect runoff and sediment produced from natural rainfall events.

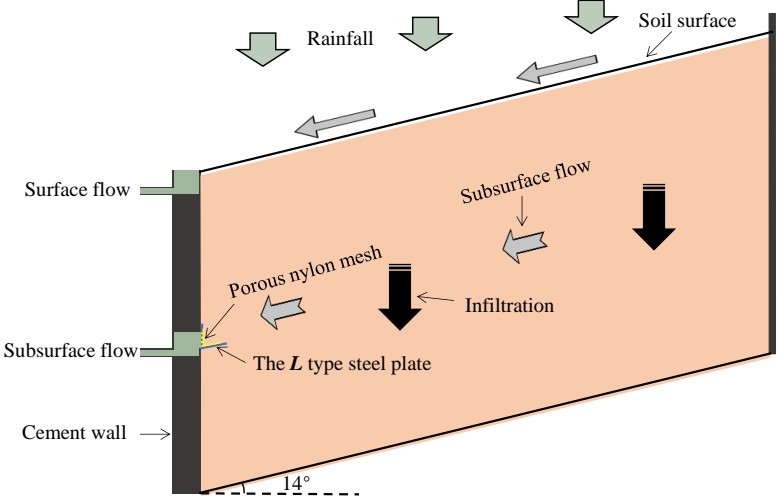




**Fig. 2** Schematic diagrams of the runoff plot and the surface-subsurface flow and sediment collection system.

*2.3 Experimental design*

In this study, the field experiment included three surface cover types: bare land, grass cover and litter cover. Bare land was used as a control treatment and weeds were manually removed from the runoff plots every two months without tillage and loosening. The grass species planted in the grass cover plot was *Paspalum natatum* Flugge, which is a quickly growing perennial grass that can be used extensively to reduce runoff and soil erosion. Grass seeds were evenly spread at a density of 20 g m$^{-2}$ in the grass cover plots after runoff plot construction. Grass growth was completely dependent on natural rainfall without human interference, such as fertilization, irrigation, reseeding, and cutting, during the study period. For the litter cover plot, a 5-cm thick layer of litter was placed on the soil surface to reduce water erosion. The litter was supplied from cutting *P. natatum* and was replenished quarterly throughout the decade.

*2.4 In situ observation*

Surface runoff, subsurface flow, and soil loss data were collected and measured for each runoff plot after each natural rainfall event during the study period. Five measurements were taken using a straightedge and the average value was used to represent the water depth. The runoff volume for each rainfall event was calculated by multiplying the container water depth by its base area, and the runoff depth was measured by dividing the runoff volume by the plot area. Then a depth profile sediment sampler was used to sample runoff samples mixed with eroded materials in a surface runoff container (Wang et al., 2016). The sediment concentration was analyzed by oven drying at 105 °C to a constant weight in the laboratory. The soil loss amount was obtained by calculating the product of the runoff volume and the corresponding sediment concentration. By





considering the hysteresis of subsurface flow, successive rainfall events were separated into two
events when the rainfall intermittent intervals exceeded 12 hours (Liu et al., 2016; Duan et al.,
2017). Rainfall event temporal profiles were automatically recorded at 1-min intervals by a
meteorological station with a resolution of 0.2 mm near the runoff plots.
*2.5 Rainfall classification based on intra-event variation*
Intra-event rainfall variation classification is important for accurately describing the time-varying
intensities that comprise a rainfall event (Dunkerley, 2021; Liu et al., 2022). The classification
refers to the overall form of a rainfall event, for example, whether the rainfall exhibits intensity
peaks in the early or late stages of an event. The application of this classification has a
considerable history. Huff (1967) introduced the classification of intra-event rainfall based on
quartiles. Specifically, according to which quarter of the event duration received the greatest
rainfall depth, the intra-event rainfall was divided into "first quartile", "second quartile", " third
quartile", and "fourth quartile" events.
In this paper, rainfall events were classified into four types of intra-event rainfall by the following
steps (Fig. 3). Firstly, the instantaneous rainfall amount and duration are divided by the total
rainfall and duration, respectively, and transformed into dimensionless parameters from 0 to 1.
Secondly, the dimensionless rainfall duration was divided into three equal parts, and the
cumulative rainfall was calculated for each equal time period. Finally, the intra-event rainfall
patterns were defined according to the location where the maximum accumulated rainfall. Equally,
rainfall events with more than 40% of the rainfall amount concentrated in the first, second and last
third periods were defined as advanced, intermediate, and delayed patterns, respectively. The
remaining events without obvious peaks and rainfall distributing uniformly over the duration were



regarded as uniform pattern (Mohamadi and Kavian, 2015; Wang et al., 2016).

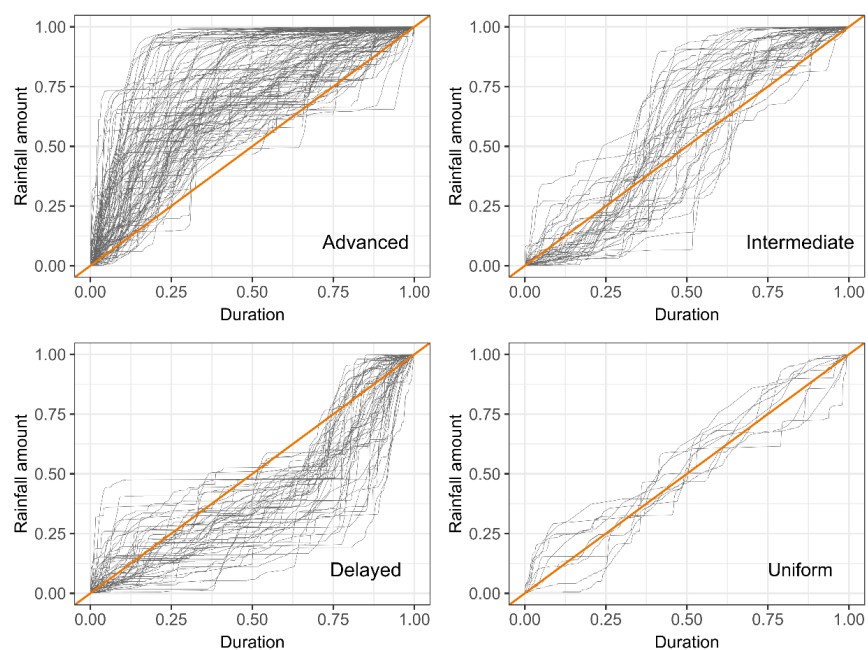


**Fig. 3** Cumulative dimensionless curves of natural rainfall events for advanced, intermediate,
delayed, and uniform patterns, respectively.
*2.6 Data analysis*
All runoff plot construction and experimental design were completed in 2000, and the in situ
observations of surface runoff, subsurface flow and sediment from natural rainfall events have
been conducted since 2001. To reduce the disturbance of plots construction on the study results,
the observation data from 2002 to 2012 were selected to analyze the surface-subsurface flow and
soil loss characteristics under different surface cover types. Three variables, surface runoff
coefficient (*ROC*), subsurface flow rate (*SSL*), and soil loss rate (*SLR*) were employed to
characterize the effect of intra-event rainfall patterns on water erosion. The *ROC* (%), *SSL* (L
mm$^{-1}$), and *SLR* (t km$^{-2}$ mm$^{-1}$) were calculated using the following equations:
$ROC = \dfrac{SRD}{PD} \times 100\%$     (1)

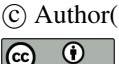



$$SSL = \frac{SFV}{PD} \qquad (2)$$
$$SLR = \frac{SLA}{A \times PD} \qquad (3)$$
where $SRD$ was the surface runoff depth (mm), $SFV$ was the subsurface flow volume (L), $SLA$ was
the sediment loss amount (g), $PD$ was the precipitation depth (mm), and $A$ was the and runoff plot
area ($m^2$).
Two-way analysis of variance (ANOVA) was used to assess the effects of intra-event rainfall
patterns, surface cover types and their interactions on $ROC$, $SSL$, and $SLR$. All statistical tests and
graphics were performed in R software v.4.1.3.
**3 Results**
*3.1 Intra-event rainfall variability*
During the observation period, 226 natural rainfall events from 2002 to 2012 that generated runoff
and soil erosion were recorded. The rainfall events were classified into four groups (advanced,
intermediate, delayed and uniform) based on rainfall profiles (Fig. 3 and Table 1). Clearly, the
prevalence of advanced pattern in the study area accounted for 48.23% of the total erosive rainfall
events. The uniform pattern had the least probability of occurrence at 5.31%. The intermediate and
delayed patterns had comparable probabilities of occurrence at 21.24% and 25.22%, respectively.
As shown in Table 1, the statistical characteristics of each intra-event pattern showed an increase
in the average duration and a decrease in the average I30 values, from the advanced to the
intermediate to the delayed to the uniform pattern. The advanced pattern was characterized by the
shortest duration (732 min) and the highest intensity (8.8 mm h⁻¹). The intermediate pattern had a
moderate duration (978 min) and moderate intensity (6.0 mm h⁻¹). The delayed pattern included
rainfall events that had the longest duration (1409 min) and moderate intensity (6.1 mm h⁻¹). The





uniform pattern was a cluster of rainfall events that had long duration (1362 min) and the lowest
intensity (2.9 mm h$^{-1}$). The average rainfall amounts for different intra-event rainfall patterns were
ranked in the order of delayed > intermediate > advanced > uniform.
**Table 1**
Rainfall eigenvalues of four intra-event rainfall patterns (IRP). D, P, and I$_{30}$ refer to rainfall
duration, depth and maximum rainfall intensity in 30 min.

| RIP | Eigenvalue | Min | Max | Mean | SD | Variation | Sum | Frequency |
|---|---|---|---|---|---|---|---|---|
| | D (min) | 27 | 4319 | 732 | 768 | 1.05 | 79761 | |
| Advanced | P (mm) | 4.6 | 130.9 | 23.2 | 19.3 | 0.83 | 2523.5 | 109 |
| | I$_{30}$ (mm h$^{-1}$) | 1.0 | 45.3 | 8.8 | 8.6 | 0.98 | | |
| | D (min) | 22 | 2972 | 978 | 726 | 0.74 | 46965 | |
| Intermediate | P (mm) | 7.1 | 72.1 | 26.2 | 17.3 | 0.66 | 1259.8 | 48 |
| | I$_{30}$ (mm h$^{-1}$) | 0.9 | 22.9 | 6.0 | 4.6 | 0.77 | | |
| | D (min) | 100 | 6191 | 1409 | 1025 | 0.73 | 80311 | |
| Delayed | P (mm) | 8.5 | 129.3 | 30.8 | 22.7 | 0.74 | 1755.6 | 57 |
| | I$_{30}$ (mm h$^{-1}$) | 1.0 | 51.2 | 6.1 | 7.3 | 1.20 | | |
| | D (min) | 426 | 2460 | 1362 | 584 | 0.43 | 16343 | |
| Uniform | P (mm) | 11.7 | 43.5 | 21.4 | 9.9 | 0.46 | 257.0 | 12 |
| | I$_{30}$ (mm h$^{-1}$) | 1.0 | 5.7 | 2.9 | 1.4 | 0.48 | | |

Fig. 4 shows the percentage and frequency distribution of four rainfall patterns under different
rainfall durations. In each duration group, there were 33, 34, 39, 66, 48, 6 rainfall events that
lasted up to 3, 3-6, 6-12, 12-24, 24-48, and more than 48 hours, respectively (Fig. 4b). This
finding suggested that the rainfall events with long duration and high amount dominated in the
study area. As shown in Fig. 4a, the percentage of the advanced pattern decreased from 81.82% to
31.33% as the rainfall duration increased, whereas that for the delayed pattern increased from 6.06%
to 50.00%. These findings indicated that the rainfall events with short duration were dominated by
advanced patterns, while the events with long duration were related to the peak intensity in the
later stage (delayed pattern).



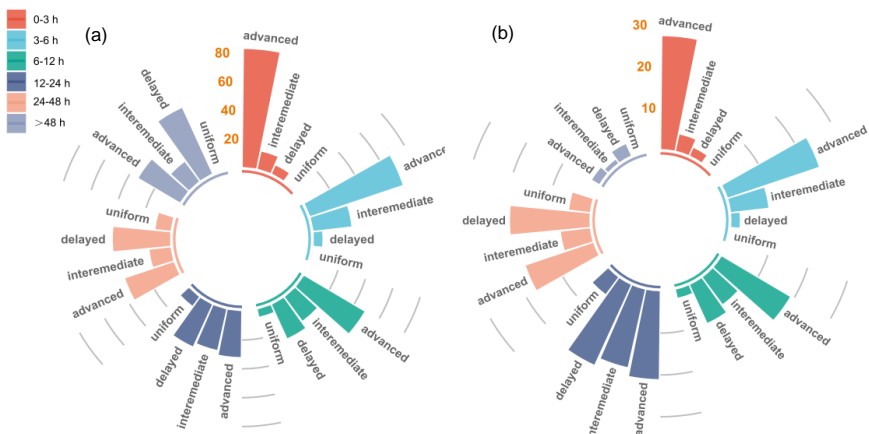


**Fig. 4** The percentage distribution (a) and frequencies (b) of intra-event rainfall patterns for
different duration groups: up to 3, 3-6, 6-12, 12-24, 24-48, and more than 48 hours.

*3.2 Surface runoff generation*

Fig. 5 shows the variation in the surface runoff coefficient (ROC) under the three surface cover

types for different intra-event rainfall patterns. Significant differences among the three surface

cover types were observed in the ROC (p<0.001). For any rainfall pattern, the ROC was

significantly higher in the bare land plot (15.83%) than in the litter cover plot (1.73%), and the

grass cover plot (1.17%). Likewise, the ROC varied significantly among the intra-event rainfall

patterns (p<0.05). For bare land, the average ROC was the highest in the advanced pattern

(22.13%), followed by the delayed pattern (17.13%), the intermediate pattern (14.91%), and the

uniform pattern (9.13%). When bare soil was covered with litter or planting grass, no obvious

differences were observed among the rainfall patterns. The values ranged from 1.61% to 1.94% in

the litter cover plot and from 1.08% to 1.23% in the grass cover plot. Meanwhile, for surface

runoff production, there were significant interactions between intra-event rainfall patterns and

surface cover types (p<0.01).

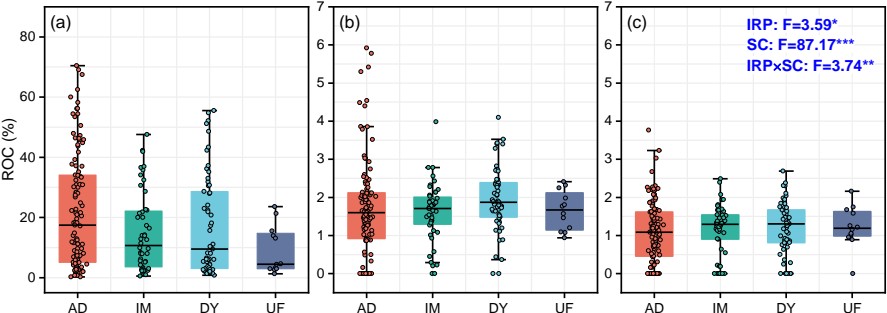

**Fig. 5** Surface runoff coefficient (ROC) under different intra-event rainfall patterns (IRP) for bare
land (a), litter cover (b) and grass cover (c) plots. Displaying F values from two-way ANOVA tests
for the effects of IRP and surface cover types (SC), and their interactions (IRP×SC) on ROC
(*$p<0.05$, **$p<0.01$ and ***$p<0.001$). AD, IM, DY, and UF refer to advanced, intermediate,
delayed, and uniform patterns, respectively.

*3.3 Subsurface flow*

The subsurface flow rates (SSLs) for the three surface cover types under different intra-event

rainfall patterns are shown in Fig. 6. The subsurface flow under the three surface cover types were

inconsistent with the surface flow. For each rainfall pattern, the lowest SSL was found in the bare

land plot (0.46 L mm$^{-1}$), followed by the grass cover plot (1.98 L mm$^{-1}$) and the litter cover plot

(2.98 L mm$^{-1}$). The subsurface flows for the litter cover and the grass cover were significantly

higher compared to the control bare land ($p<0.001$). From the rainfall patterns, the subsurface flow

rate for the bare land decreased in the order of delayed pattern (0.63 L mm$^{-1}$) > uniform pattern

(0.50 L mm$^{-1}$) > intermediate pattern (0.44 L mm$^{-1}$) > advanced pattern (0.29 L mm$^{-1}$). For the

litter cover, the SSL values differed slightly and ranged between 2.93 and 3.13 L mm$^{-1}$ under

different intra-event rainfall patterns. The SSL for grass cover was lower than that for litter cover,

and the SSL values decreased in the order of delayed pattern (2.44 L mm$^{-1}$) > intermediate pattern

(2.22 L mm$^{-1}$) > advanced pattern (1.87 L mm$^{-1}$) > uniform pattern (1.41 L mm$^{-1}$).



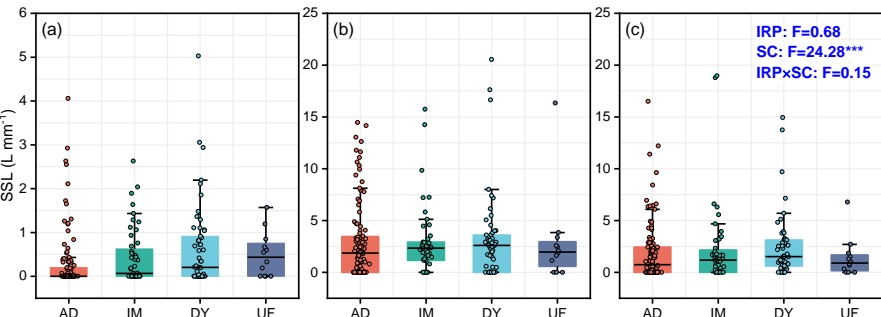

**Fig. 6** Subsurface flow rate (SSL) under the four intra-event rainfall patterns (IRP) for bare land
(a), litter cover (b) and grass cover (c) plots. Displaying F values from two-way ANOVA tests for
the effects of IRP, surface cover types (SC), and their interactions (IRP×SC) on SSL (*$p < 0.05$,
**$p < 0.01$ and ***$p < 0.001$). AD, IM, DY, and UF refer to advanced, intermediate, delayed, and
uniform patterns, respectively.

*3.4 Sediment yield*

As shown in Fig. 7, the soil loss rate (SLR) of the litter cover and grass cover decreased drastically

compared to that of the bare land. The SLR for the bare land was the highest (2.18 t km$^{-2}$), being

and 253 times greater than the SLRs for the litter cover and grass cover, respectively. The

results were consistent with the surface flow of the three surface cover types. Significant

differences in the SLR were identified between the intra-event rainfall patterns ($p < 0.001$). The

highest SLR for bare land was observed in the advanced pattern (5.14 t km$^{-2}$), with values that

were 2.52, 3.68 and 39.78 times more than those of delayed, intermediate, and uniform patterns,

respectively. Compared with the bare land, the SLR differences among the rainfall patterns

decreased sharply for the litter cover and grass cover, with values ranging from 0.009 to 0.013 t

km$^{-2}$ and from 0.005 to 0.011 t km$^{-2}$, respectively. Consistent with the surface runoff, significant

interactions between intra-event rainfall patterns and surface cover types were observed in the soil

loss rate ($p < 0.001$).



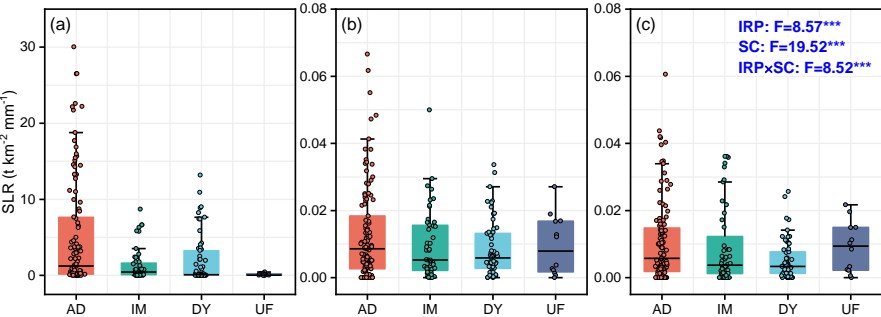

**Fig. 7** Soil loss rate (SLR) under different intra-event rainfall patterns (IRP) for bare land (a), litter
cover (b) and grass cover (c) plots. Displaying F values from two-way ANOVA tests for the effects
of IRP, surface cover types (SC), and their interactions (IRP×SC) on SLR (*p<0.05, **p<0.01 and
***p<0.001). AD, IM, DY, and UF refer to advanced, intermediate, delayed, and uniform patterns,
respectively.

**4 Discussion**

*4.1 Effects of intra-event rainfall variation on runoff generation and soil loss*

In addition to inter-event rainfall variation, the intra-event variation significantly alters rainfall

infiltration, runoff generation and erosion processes (Dunkerley, 2021; An et al., 2022).

Classification based on rainfall profiles is important for describing intra-event rainfall variation.

This classification is the basis for accurately elucidating the mechanisms of rainfall characteristics

on water erosion (Parsons and Stone, 2006; Dunkerley, 2021). In this study, according to the

occurrence period of the maximum rainfall amount, four intra-event rainfall types were classified

based on 1-minute interval rainfall data from 12 consecutive years of long-term in situ

observations (262 events). As shown in Table 1, the advanced and delayed patterns were

characterized by short duration/heavy intensity and long duration/high amount, respectively. These

two rainfall patterns were the dominant events in the study area, accounting for 73.45% of the

total erosive rainfall events. The study area has a subtropical humid monsoon climate with two

typical rainfall types of long-duration plum rains in spring and short-duration storms in summer

(Liu et al., 2016; Duan et al., 2017). The results indicated that the intra-event rainfall types





obtained in this paper were consistent with the actual situation of rainfall characteristics in the
study area. The natural rainfall data selected in this paper exhibited good typicality and
representativeness, and the method for classifying intra-event rainfall types was relatively reliable.
However, the intra-storm variations during natural rainfall processes are extremely complex,
which makes them very difficult to quantize (Dunkerley, 2021; Liu et al., 2022). The classification
method used in this study described only one aspect of inter-event rainfall variation, and did not
completely capture all of the properties. Therefore, the development of an available and excellent
index system to quantify intra-event rainfall variability remains a topic requiring an in-depth study
in the future.
Figs. 5 and 7 indicate that the surface runoff coefficients and soil loss rates from intra-event
variation patterns were 1.63 to 2.42 times and 15.79 to 39.78 times greater than those from the
uniform pattern, respectively. For soil loss, the results were consistent with previous studies based
on rainfall simulation (Flanagan et al., 1988; Parsons and Stone, 2006; An et al, 2014; Wang et al.,
2017). The studies showed that varying-intensity storms yielded more soil loss than
constant-intensity storms. Similar to simulated rainfall, the intra-event variability for natural
rainfall events played important roles in slope soil erosion. However, studies on the effects of
intra-event rainfall patterns on surface flows had different results. Dunkerley (2012) determined
that uniform events of unvarying intensity yielded the lowest total runoff, the lowest peak runoff
rate and the lowest runoff ratio. The researches found that that varying intensity rainstorms did not
affect total runoff or infiltration (An et al., 2014; Wang et al., 2017). The reasons for the above
results difference were mainly related to factors such as antecedent soil water content, soil types
and topography (Frauenfeld and Truman, 2004; Parsons and Stone, 2006; Alavinia et al., 2019).





The uncertainty of intra-event rainfall variation on runoff generation was higher than that of
sediment yield. In addition, the variation coefficients of soil loss (1.73) were higher than those of
surface runoff (0.93) on bare land when the rainfall patterns changed (Table 2). The results
indicated that sediment yields were more sensitive to intra-event rainfall variation than surface
runoff.
**Table 2**
The variation coefficients of surface runoff coefficient (ROC), subsurface flow rate (SSL) and soil
loss rate (SLR) induced by different intra-event rainfall patterns (IRP) and surface cover types
(SC).

| Variation factor | Fixed factor | ROC | SSL | SLR |
|---|---|---|---|---|
| IRP | Bare land | 0.93 | 2.16 | 1.73 |
| | Litter cover | 0.61 | 1.45 | 1.10 |
| | Grass cover | 0.65 | 2.47 | 1.44 |
| SC | advanced | 1.78 | 2.10 | 2.81 |
| | intermediate | 1.68 | 2.00 | 3.00 |
| | delayed | 1.78 | 2.63 | 3.07 |
| | uniform | 1.45 | 1.79 | 2.01 |

As shown in Fig. 8, the power function relationship between surface runoff and soil loss was
influenced by intra-event rainfall variation. The soil loss rate ranged in the order of advanced
pattern > delayed pattern > intermediate pattern > uniform pattern under the same surface runoff
depth. For bare land, the highest surface runoff and soil loss were found in the advanced pattern,
which were 1.29 to 2.42 times and 2.52 to 39.78 times higher than those in the other three patterns
(Fig. 5a and 7a). The advanced pattern was the main type for surface flow and soil loss in the bare
land, with contribution rates of 57.24% and 75.17%, respectively (Fig. 9). All of the above results
indicated that the events with more rainfall concentrated in the early stages were more favourable
for surface flow and soil loss. The results were similar to the research of Römkens et al. (2001),
who found that the falling pattern caused more soil loss than the rising pattern. The findings
contrasted with those of An et al. (2014), Mohamad and Kavian, (2015), and Wang et al. (2016),





who reported that the delayed pattern yielded the most soil loss under the same average intensity.
The reason for this phenomenon was possibly the complexity of natural rainfall and the
differences in soil type. In this study, the soil type was clayey red loam with relatively low
permeability. Due to the initially high intensity in the advanced pattern, interrill erosion and
physical crusts were rapidly formed, resulting in decreasing soil infiltration rate and an increase
surface runoff (Liu et al., 2011). The concentrated flow was generated early because of excess
infiltration for the advanced pattern with short duration and heavy intensity (Table 1). Moreover,
in the initial period of rainfall, rapid infiltration quickly increased topsoil moisture and reduced
soil aggregate stability, resulting in increasing the soil erodibility (Le Bissonnais, 2010). The
combined effect of increased surface runoff and decreased soil resistance to erosion inevitably
increased soil erosion. These processes were closely related to antecedent soil water content, soil
texture and soil configuration (Frauenfeld and Truman, 2004; Alavinia et al., 2019). Compared to
the other three patterns, the uniform events with long duration and low intensity led to surface soil
with lower splash erosion and slower seal formation, higher soil infiltration rate and subsequently
less surface runoff and soil loss (Dunkerley, 2012).

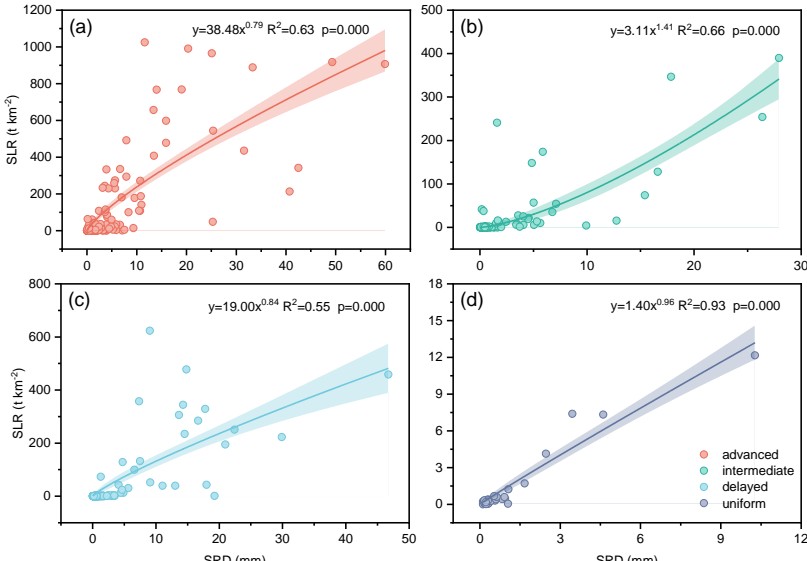


**Fig. 8** The relationship between surface runoff depth (SRD) and soil loss amount (SLR) under
different intra-event rainfall patterns.

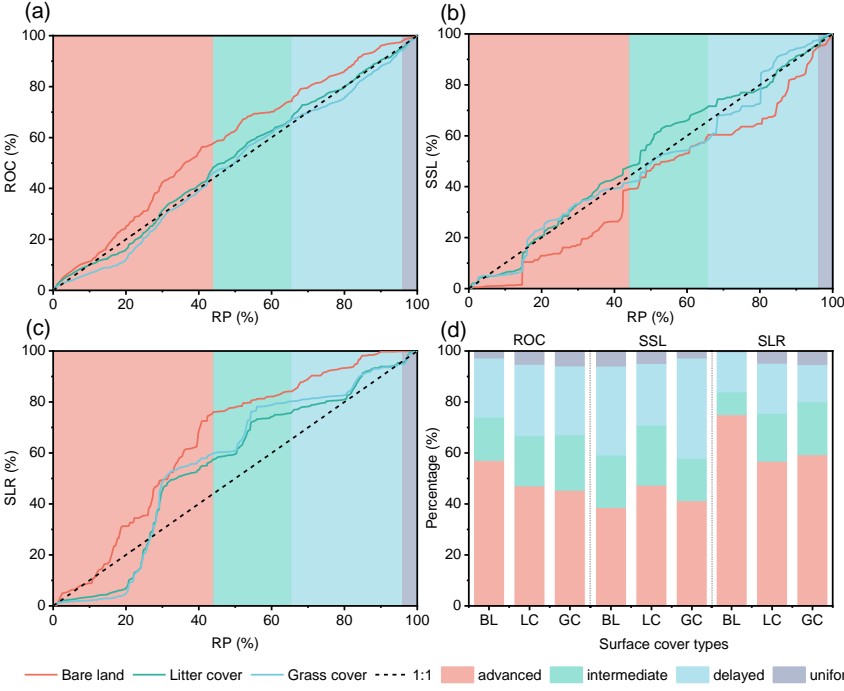


**Fig. 9** The relationships between rainfall depth (RP) cumulative percentage and surface runoff
coefficient (ROC), subsurface flow (SSL) and soil loss rate (SLR) cumulative percentages under
different surface cover types. BL, bare land; LC, litter cover; GC, grass cover.





Relative to surface runoff and soil loss, the delayed pattern had the greatest average subsurface
flow, which was 1.26, 1.43, and 2.17 times more than the flow characteristics of the uniform,
intermediate, and advanced patterns, respectively (Fig. 6). This phenomenon was highly related to
the rainfall characteristics and slope runoff generation mechanisms of the above intra-event
rainfall patterns (Wang et al., 2016; Liu et al., 2016). Table 1 clearly shows that the delayed and
uniform patterns were characterized by long-duration and heavy-intensity rainfall. The duration
groups of 12-24 h, 24-48 h and >48 h in the delayed pattern accounted for 31.82%, 39.58% and 50%
of the total rainfall events, respectively. The uniform pattern was only distributed in 6-12 h, 12-24
h, and 24-48 h groups (Fig. 4). For the delayed and uniform patterns, rainfall intensity in the early
stage was lower than topsoil infiltration rates. Most of the rainfall infiltrated and accumulated in
the topsoil layer because of the compact subsoil (Ma et al., 2022a). The subsurface flow under
constant rain for a long time formed easily owing to excess storage and the lateral slope when
topsoil moisture reached saturation. In subtropical humid monsoon climate zones, subsurface flow
could even exceed surface flow and become the primary cause of water loss in some exceptional
rainfall events (Liu et al., 2016; Duan et al., 2017). Notably, high subsurface flow caused an
increasing in the soil erosion risk (An et al., 2021). As a result, in terms of soil hydrology and
erosion process, in addition to advanced rainfall with a short duration and high intensity, more
attention should be paid to delayed and uniform rainfall events with long duration.
*4.2 Surface cover regulating the impact of intra-event rainfall variation on water erosion*
Vegetation effectively increases rainfall infiltration and reduces surface runoff and soil erosion
(Han et al., 2021; Duan et al., 2022). For each rainfall pattern, long-term continuous in situ
observations showed that litter cover and grass cover significantly reduced surface runoff and soil





erosion and increased subsurface flow compared to bare land (Fig. 5-7). Reductions in surface
runoff and sediment ranged from 88.01 to 91.69% and from 97.80 to 97.95%, respectively, while
subsurface flow was 3.55 to 5.92 times greater in surface covered plots (Table 3). The effects of
litter cover on reducing surface runoff and soil loss were comparable to those of grass cover. This
similarity could be because the two cover types had similar coverage closely relating to surface
runoff and sediment loss (Hou et al., 2020). The results confirmed that surface cover with plant
litter or grass was effective in reducing surface runoff and erosion, and the influences were not
affected by intra-event rainfall variation.
**Table 3**
Average surface runoff coefficient (ROC), subsurface flow rate (SSL) and soil loss rate (SLR)
reduction benefit under different intra-event rainfall patterns (IRP).

| IRP | ROC (%) | | SSL (%) | | SLR (%) | |
|---|---|---|---|---|---|---|
| | Litter cover | Grass cover | Litter cover | Grass cover | Litter cover | Grass cover |
| Advanced | 92.29 | 95.14 | -915.44 | -546.63 | 99.75 | 99.79 |
| Intermediated | 89.22 | 92.16 | -563.62 | -403.20 | 99.31 | 99.34 |
| Delayed | 88.69 | 92.91 | -396.94 | -286.90 | 99.58 | 99.73 |
| Uniform | 81.84 | 86.55 | -492.89 | -184.46 | 92.57 | 92.93 |
| Average | 88.01 | 91.69 | -592.22 | -355.30 | 97.80 | 97.95 |

The effects of intra-event rainfall variation on surface runoff and sediment loss were strongly
influenced by the surface cover. Compared to bare land, the contributions of the advanced pattern
to surface flow and soil loss for the surface cover decreased by 9.97-11.69% and 15.68-19.31%,
respectively (Fig. 9). The statistical results showed that the variation coefficients of surface runoff
and soil erosion for bare land were the highest, and those were more than those for litter cover and
grass cover when the rainfall patterns changed (Table 2). The above results indicated that surface
cover weakened the impacts of intra-event rainfall variation on surface flow and soil loss. This
phenomenon occurred because surface cover effectively reduced splash erosion from rainfall
concentration, and led to surface soils without sealing formation, a higher soil infiltration rate and





subsequently lower runoff and sediment loss (Wei et al., 2007; Liu et al., 2016; Wang et al., 2020).
In addition, the effective buffer layer on the soil surface increased the surface roughness, and
delayed the overland flow velocity, thereby reducing the scouring ability of surface runoff on soil
(Shi et al., 2012; Fu et al., 2020; Liu et al., 2020). The improvement in the soil anti-erosion ability
was another important reason for this phenomenon (Wang et al., 2020; Ma et al., 2022b).
Different measures covering the soil surface had varying near-surface characteristics and effects
on soil properties along the profile, resulting in disinct soil hydrological responses. The increasing
benefit of subsurface flow for litter cover for each rainfall pattern was 1.38 to 2.67 times greater
than that of grass cover (Table 3). Due to no difference in surface runoff, more rainfall was
converted to soil water storage and deeper infiltration under grass cover than under litter cover for
constant rainfall. For long-term coverage of the soil surface, the plant litter was buried by soil
particles from water and wind erosion (Hewins et al., 2017). The incorporation of plant litter into
the topsoil layer actively impacted soil hydraulic properties such as bulk density, soil infiltration,
and saturated hydraulic conductivity (Jordán et al., 2010; Wang et al., 2020). Rainfall infiltration
tended to take the form of matrix infiltration, with large amounts of rainfall being stored in the
topsoil layer. Because of loose top- compact bottom soil configuration, the topsoil was easily
saturated and generated subsurface flow, but rainfall was more difficult to convert into deep soil
moisture (Fig. 10b). For grass cover, rainfall infiltration was greater in the form of preferential
flow with many root channels (Guo et al., 2019). Before topsoil saturation, rainfall was highly
susceptible to rapid infiltration into the subsoil by preferential flow, leading to little subsurface
flow and large water storage in the deeper layers (Fig. 10c).



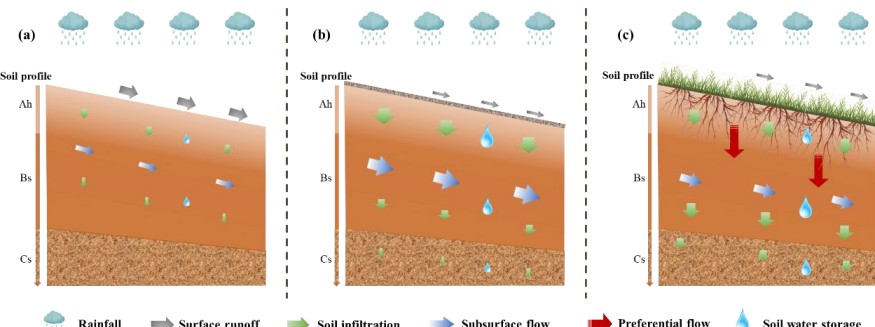

**Fig. 10** Conceptual diagram of the soil hydrological processes under the bare land (a), litter cover (c) and grass cover slopes.

The increasing effects of surface cover on subsurface flow varied among different rainfall patterns.

For litter cover and grass cover, the highest increases were observed in the advanced pattern at

915.44% and 546.63%, respectively, which were significantly higher than those in the

intermediate, uniform, and delayed patterns (Table 3). As shown in Fig. 9, surface cover increased

the contribution rate of the advanced pattern to the subsurface flow from 38.76 to 47.54%. The

reason for this phenomenon was that the surface cover increased the soil infiltration capacity and

prevented the formation of surface crusts, increasing the amount of rainfall that was concentrated

earlier to infiltrate the soil. Concerning subsurface flow variation, the coefficient of variation for

litter cover was much smaller than the coefficients for bare land and grass cover (Table 2). The

results illustrated how the effects of intra-event rainfall variability on subsurface flow were easily

masked by litter cover. This phenomenon occurred due to the strong water absorption of the litter

layer, which was an important source for steady water infiltration (Darboux et al., 2002). For the

grass cover, the water absorption capacities of the aboveground parts were relatively weak, and

rainfall infiltration and subsurface flow were more susceptible to intra-event rainfall changes. This

response was particularly observed for events with more rainfall concentrated in the early stages,

which tended to form preferential flows.



*4.3 Implications and further scopes*


Human disturbances played a crucial role in surface runoff and soil erosion intensification. For
example, serious soil disturbance induced by large scale mechanical excavation resulted in
enormous regions of bare soil during agricultural land use (Niu et al., 2021). The inter- and
intra-event variations were important consequences of rainfall changes with global warming
(Dunkerley, 2021b; An et al., 2022). The increase in extreme rainfall frequency and intensity was
the most typical form of expression, and finally water erosion risk was aggravated (Nie et al.,
2020; Shenoy et al., 2022). In this paper, based on long-term in situ observations, the two
measures of surface cover showed very good stability in reducing surface runoff and sediment,
regardless of the inter- and intra-event rainfall variation. In addition, the differences in surface
runoff and soil loss from the inter- and intra-event rainfall changes were weakened by litter cover
or grass cover. The results showed that covering bare soil with plant litter or planted grass can
effectively mitigate the water erosion risk caused by climate change and unreasonable human
activities.
The increased risk of drought frequency, duration, and intensity was another important issue
arising from global climate change and anthropogenic impacts. Droughts have severe
environmental and socio-economic influences, especially in countries relying on rain-fed
agricultural production (Sternberg, 2011; Chiang et al., 2021). Storing more precipitation in soil
during the flood season is an effective way to address this hazard, especially to increase deep soil
water storage. Deep soil moisture provided a potential water source for crops utilization during the
dry season (Wu et al., 2021). In this study, surface runoff from the slope was comparable under the
two cover measures, while subsurface flow was greater under litter cover than grass cover. In




other words, the grass cover resulted in more soil water storage and deeper permeation than the
litter cover. Moreover, soil water storage was mainly in the topsoil layer for litter cover and in the
subsoil and deeper layers for grass cover. Therefore, to improve soil, crop and water productivity
under rainfed hill ecosystems, there is a great need to adapt single mulching to diversified
mulching measures (Ngangom et al., 2020). For example, double mulching technology involving
plant litter and planted grass can increase shallow and deep soil water storage while reducing
surface runoff and erosion, and mitigate the hazards of agricultural production caused by extreme
climate.
**5 Conclusion**
In this study, 262 natural rainfall events were classified into four intra-event rainfall patterns
(advanced, intermediate, delayed, and uniform) over 11 consecutive years in the red soil region of
China. Three surface cover types including bare land, litter cover and grass cover were selected to
study the response of surface-subsurface flow and soil loss to intra-event rainfall variation. The
advanced pattern was the most frequent rainfall event that had the shortest duration and the
highest intensity. The intermediate pattern was represented by moderate duration and moderate
intensity. The delayed pattern involved the longest duration and highest depth. The uniform
pattern was characterized by long duration and the lowest intensity and frequency.
Surface runoff and soil loss in the advance pattern were highest, followed by the delayed,
intermediate, and uniform patterns. Sediment yields were more sensitive to intra-event rainfall
variation compared to surface runoff. However, the subsurface flow was in the order of delayed
pattern > uniform pattern > intermediate pattern > advanced pattern. For all rainfall patterns, bare
land had the highest surface runoff and soil loss. Surface cover significantly reduced surface



runoff and soil erosion by 88.01 to 91.69% and 97.80 to 97.95%, respectively, while subsurface
flow increased from 3.55 to 5.92 times. The reduction benefits of litter cover were comparable to
those of grass cover. However, compared to bare land, the increasing benefit of subsurface flow
for litter cover for each rainfall pattern ranged from 1.38 to 2.67 times that of grass cover.
Moreover, surface cover can weaken the influences of intra-event rainfall variation on
surface-subsurface flow and soil loss. These findings could enhance the understanding of the
impacts of rainfall changes at inter- and intra- event scales on key surface processes such as
surface-subsurface flow and soil erosion, and provide a basis for optimizing surface cover
measures to effectively respond to extreme disasters caused by global climate change.
**Data availability**
The data that support the findings of this study are available from the corresponding author upon
request.
**Author contributions**
Jian Duan was responsible for the data investigation, analysis, and methodology and completed
the original draft of the article. Haijin Zheng and Yaojun Liu conceptualized this research,
including conceptualization and methodology, and participated in the review and editing of the
writing. Minghao Mo contributed to the data investigation and the review and editing of the
writing. Yuejun Song and Jie Yang were involved in writing review and editing.
**Competing interests**
The contact author has declared that none of the authors has any competing interests.
**Acknowledgements**
This research was supported by the National Natural Science Foundation of China (42107378,



41877084, 42067020) and Science and Technology Project of Jiangxi Province (20212AEI91011,
2022KSG01001). We would like to acknowledgements the anonymous reviewers, associate editor
and editor for their valuable comments and kind assistance on this manuscript.

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
