# Peer review of "field observations"

_Hydrology and Earth System Sciences, 2023_

## Author Comment (AC1)

We would like to thank David Dunkerley for reviewing and commenting on our paper. We find this discussion very interesting and feel that it will enrich the final version of the paper. You will find your comments in **black**, while our responses are given in **blue** and any citation how we suggest to revise the text in the revised manuscript in **red**.

**Reviewer #1:**

This paper analyses runoff and soil loss data collected from several experimental plots, with data covering about a decade of observations under natural rainfall. I would first commend the authors for having adopted field study under natural rainfall, rather than the generally inappropriate reliance on "rainfall" simulation (which typically uses unrealistic, fixed, high intensities - often with little or no attempt at offering a reasoned basis for this, and typically neglecting intensity profile altogether). I think that observations made under natural rainfall are arguably far more informative and valuable than "rainfall" simulation studies that neglect the kinds of issues that are discussed in the current manuscript.

**Response:** Thank you for the nice comments on the significance of this study. We took them all into account in detail and respond to each below:

The use of four classes of intra-event rainfall pattern (advanced, intermediate, delayed, and uniform) is not unusual, and does provide some basis for identifying and categorising the intra-event rainfall intensity variation. However, this classification provides no indication of what the actual intensities during an event were. Indeed, it is a limitation of the current manuscript that remarkably little is actually said about the rainfall intensities, beyond reporting of the mean event intensity (I presume) for the advanced, delayed, etc. (e.g. lines 234-236). I think that it would have been helpful and appropriate for the paper to report additional detail concerning intensity - perhaps using measures like I5, I15, I30, I60, and so on. In the absence of this, we cannot really understand whether, for instance, the 'advanced' event type had different peak intensities than the other classes, and for how long those peak intensities might have lasted. We cannot see whether there was more intensity variation in advanced than in delayed, for instance, and it is well-known that short-term intensity peaks can exert critical influences on soil erosion (e.g. see for instance Dunkerley, D. L. (2019). Rainfall intensity bursts and the erosion of soils: an analysis highlighting the need for high temporal resolution rainfall data for research under current and future climates. Earth Surf. Dynam., 7(2), 345-360. doi:10.5194/esurf-7-345-2019).

**Response:** Thank you for the insightful suggestions. We agree to provide more information about intensities during an event. As suggested by the reviewer, short-term intensity peaks can exert critical influences on soil erosion. It is well-known that the maximum 30-min intensity of rainfall events (I30) had a significantly positive relationship with the soil erosion amount. In order to explore the peak intensity differences among the four patterns, I30 and accumulative rainfall percent for each 1/3

period were provided in Table 1. As shown in Table 1, the I30 of advanced patterns was more than the other patterns. The results were the main reason for the highest soil loss amount in advanced patterns.

**Suggested revision in the text:**

**Table 1**

Rainfall eigenvalues of four intra-event rainfall patterns (IRP). D, P, and $I_{30}$ refer to rainfall duration, depth and maximum rainfall intensity in 30 min.

| RIP | Sample size | Rainfall percent for each 1/3 period (%) | | | D (min) | | | P (mm) | | | $I_{30}$ (mm h$^{-1}$) | | |
|---|---|---|---|---|---|---|---|---|---|---|---|---|---|
| | | 0~1/3 | 1/3~2/3 | 2/3~1 | Min | Max | Mean | Min | Max | Mean | Min | Max | Mean |
| Advanced | 109 | 74 | 15 | 11 | 27 | 4319 | 732 | 4.6 | 130.9 | 23.2 | 1.0 | 45.3 | 8.8 |
| Intermediate | 48 | 22 | 64 | 12 | 22 | 2972 | 978 | 7.1 | 72.1 | 26.2 | 0.9 | 22.9 | 6.0 |
| Delayed | 57 | 11 | 10 | 79 | 100 | 6191 | 1409 | 8.5 | 129.3 | 30.8 | 1.0 | 51.2 | 6.1 |
| Uniform | 12 | 33 | 36 | 31 | 426 | 2460 | 1362 | 11.7 | 43.5 | 21.4 | 1.0 | 5.7 | 2.9 |

I would like to have seen in the paper more information on rainfall continuity or intermittency also, as breaks in rainfall can be critical in allowing overland flow to slow and for soil to be re-deposited, for ephemeral surface ponding to dissipate, for soil infiltrability to partially recover, and so on. Much is known about all of this, and could have been considered in the manuscript. Many rainfall events reported in the manuscript have durations of more than 24 hours, and up to 48 hours. Was rainfall actually continuous through these long durations, or were there breaks (cessations) in rainfall? The authors might also say something about when the rain occurred, especially for the 'advanced' type. These might for instance have been late afternoon convective events, whilst the 'delayed' type might have been long, overnight falls. The timing, diurnal or nocturnal, would influence evaporation rates, especially during any temporary breaks in rainfall. This is too often ignored when rainfall data are reported simply in terms of event amounts and intensities.

**Response:** Thank you for the valuable suggestion. The discussion on rainfall intermittency contributed to understand the underlying mechanisms of different types of intra-event variation on surface-subsurface flow and soil loss. As suggested by the reviewer, we have added the discussion on the effect of rain intermittency on runoff generation and soil erosion under different intra-event rainfall patterns. We revised the text accordingly as follows:

**Suggested revision in the text:** The characteristic intermittency of rainfall includes temporary cessations (hiatuses), as well as periods of very low intensity within more intense events (Dunkerley, 2018; 2021). For instance, Figure 3 clearly shows that there is ubiquitous intermittency in most natural rainfall events, especially for long-duration rainfall events. The advanced pattern events might have been late afternoon convective events. Intense rainfall intensity was concentrated in the early period, while intermittency often occurs in the later periods (Figure 3). The intense intensity in the advanced pattern tended to induce strong runoff scour in the early periods, which

greatly contributed to the development of concentrated flow and produced more intense soil erosion. In addition, early intense rainfall may also induce more splash erosion, which also provides abundant loose soil particles and thus leads to greater sediment production capacity. Conversely, the delayed pattern might have been overnight falls with long time. Most of rainfall interval occurred in the early and middle stages (Figure 3), and therefore the apparent infiltration rate expands, which greatly increased the time to first runoff and reduced surface runoff and soil erosion in the early stages. In addition, the subsurface flow was increased due to the increase of infiltration rate and intense rainfall intensity in later periods (Section 4.1).

Dunkerley, D., 2018. How is overland flow produced under intermittent rain? An analysis using plot-scale rainfall simulation on dryland soils. Journal of Hydrology, 556, 119–130. https://doi.org/10.1016/j.jhydrol.2017.11.003

Dunkerley, D., 2021. Intermittency of rainfall at sub-daily timescales: New quantitative indices based on the number, duration, and sequencing of interruptions to rainfall. Atmospheric Research, 253, 105475. https://doi.org/10.1016/j.atmosres.2021.105475

I was not entirely convinced that simply classifying rainfall events as 'advanced', 'delayed', etc. is sufficient. I think that the authors recognised this too. Clearly, the depth and intensity of the events differed widely, and this is where measures such as I30 or a related index might have been useful. In my own experimental studies, designed primarily to understand how intensity profile affects infiltration and runoff, all events, regardless of intensity profile, had the same depth, duration, and average intensity. An example is Dunkerley, D. (2012). Effects of rainfall intensity fluctuations on infiltration and runoff: rainfall simulation on dryland soils, Fowlers Gap, Australia. Hydrological Processes, 26(15), 2211-2224. doi:10.1002/hyp.8317. In that work, all events lasted 90 minutes and delivered 15 mm of rainfall at an average rainfall rate of 10 mm h$^{-1}$. This ability to hold depth and duration constant experimentally isolates, at least to some extent, the intensity profile itself as the factor than could account for differences in plot infiltration and runoff. But this is not possible to do when using natural rainfall. Therefore, event durations were different among the intensity profiles analysed by the authors of the present manuscript. Were the 'advanced' events primarily different in their effects on the soil plots from the 'delayed' events in terms of their intensity profile, or their duration, or their peak intensity, or the duration of intensity exceeding, say, 10 mm h$^{-1}$, or some other factor or factors? I think that the present manuscript leaves this unresolved. Perhaps issues of this kind could be addressed in future work by the authors. I well understand that not all questions can be addressed in a single manuscript of manageable length. It is clear from Figure 3 for instance that the 'advanced' events are much more variable in their intensity profiles than any of the other classes. The spread of event characteristics declines in the sequence advanced > delayed > intermediate > uniform. This suggests, for instance, that it might be advantageous to consider some sub-categories within at least the 'advanced' class (and perhaps also in the 'delayed'). A suitable measure might be something like 'time to peak intensity', say. It is evident from Figure 3 that some 'advanced' events are more

advanced than others (i.e., that their rainfall arrives much earlier). This might be worth considering in future work.

**Response:** Thank you for the valuable suggestion. Rainfall simulation is a widely used research tool with the advantage of controlling single-factor variables. It provides control and facilitates the complex task of building an understanding of landsurface processes from field or laboratory experiments. Dunkerley (2012) simulated fourteen rainfall events each involved more than 5000 changes of intensity and included multipeak events with a 25 mm h$^{-1}$ peak of intensity early in the event or late in the event and an event that included a temporary cessation of rain. The depth and mean intensity of all rainfall were 15 mm and 10 mm h$^{-1}$, respectively. The results demonstrated that event profile (peak intensity in which period) exerted an important effect on infiltration and runoff. However, there is a discrepancy between simulated rainfall and natural rainfall conditions. Rainfall event profile simulation experiments may not adequately capture the complexity of natural rainfall processes, and the simulation rainfall derived results need to be verified by more field natural rainfall experiments.

Rainfall intensity profiles have an extremely important and complex influences on infiltration, runoff generation, soil erosion and related landsurface processes (Dunkerley, 2021). As the reviewer said, the effects and its underlying mechanism of intra-event rainfall pattern on runoff and soil erosion were addressed difficulty in a single manuscript of manageable length. This also maked a bloated and unfocused paper. We are very grateful for the valuable comments given by the reviewer, which have given us a new perspective to analyze in depth the mechanisms of intra-event rainfall variations on soil erosion. In the future work, a series of indicators were proposed to describe the features of rainfall intensity fluctuation. For example, the fraction of duration in low-intensity zone (intermittency), the fraction of rainfall amount in high-intensity zone, the relative amplitude of rainfall intensity, and the relative number of rainfall peaks, etc. The effects of rainfall intensity fluctuation on surface-subsurface flow and soil loss are quantified.

The authors also neglect, as is often the case in such studies, the time sequence of rainfall events. This sequence influences the critical antecedent wetness (or dryness) of the plot soils prior to the start of an event. Events that begin soon after a prior event, with soils that are already partially wetted-up, are likely to have infiltration and runoff characteristics that are different to those seen when an event begins after weeks without rain, say. The literature is full of findings on such effects, and they really should not be ignored. In my own experimental work, referred to above, each experiment was made on a fresh runoff plot, and none had received any rain for some weeks, such that all plots could be regarded as having the same initial condition - 'dry'. So I would speculate that an 'advanced' event soon after a prior event would show very different runoff behavior than one on dry soil, say. And it makes limited sense to compare what happens in an 'advanced' event on wet soil with what happens on a 'delayed' event on dry soil - because it is not only the class of event that is different, but the antecedent soil wetness. The two effects are then confounded in any statistical analysis. Clearly

this cannot be controlled for in work under natural rainfall, but nevertheless, antecedent conditions and their influence have to be borne in mind, and their possible effect analysed as well as can be managed. One could, perhaps, analyse as a test case only rainfall events that occurred after at least 5 days without rain, or some such criterion.

**Response:** We totally agree with your suggestion. The significant and complex role of soil moisture content on soil hydrology and erosion processes is recognized by many studies. Soil under different antecedent wetness or dryness had different hydrology and erosion response.

The antecedent soil moisture was not discussed in this paper for the following reasons. Firstly, unlike simulated rainfall, the antecedent soil moisture before each runoff event cannot be consistent under field observations. This is also a major feature of runoff and sediment analysis from natural rainfall events. Moreover, due to high spatiotemporal variation of soil moisture, antecedent soil moisture had not been mainly considered as a single factor during the most studies about the impacts of natural rainfall patterns of water erosion (Wei et al., 2007; Fang et al., 2012; Liu et al., 2016; Feng et al., 2020; Yang et al., 2022; Liang et al., 2023). Thirdly, rainfall events that occurred after at least 5 days without rain were selected to analyze. This is a good way to solve the error brought by the difference of antecedent soil moisture under the same land use type. However, surface cover on bare soil had a great influence on soil moisture redistribution under natural rainfall events. In this paper, soil water distribution in the three plots was significantly different under the same rainfall conditions. Therefore, even based on the above criteria, it may not be possible to obtain the runoff events with relatively consistent antecedent soil moisture.

It is an important solution to improving the credibility of natural rainfall experiment by increasing the sample numbers (rainfall events) as much as possible. This also reflects the high value of long-term in situ field observations of natural rainfall. In other words, with a large number of samples (rainfall events), the importance of intra-event rainfall variability in soil erosion is better demonstrated. In this paper, 262 rainfall events were obtained by the method of long-term in situ observation (11 consecutive years). Although it cannot explain the role of antecedent soil moisture in runoff generation and sediment loss, it truly reflects the influence of natural rainfall patterns on water erosion. The results showed that natural rainfall patterns have significant effects on surface-subsurface flow and soil loss. It fully indicates that the results are credible.

The combined effects of natural rainfall patterns and antecedent soil moisture on runoff generation and soil loss were the focus of our future research. To obtain the temporal dynamics of soil moisture, automatic monitoring instruments were installed along the slope. The influences of rainfall classification on water erosion under different antecedent soil moisture were detailly investigated and discussed.

The manuscript is generally clear and easy to read. There are minor errors that could be corrected easily. The authors variously report that they analysed 226 rainfall events (line 225) or 262 events (lines 321, 504). I am not sure which number is correct. I doubt that the plot walls were actually 100 cm (1 m) tall - as stated in line 148. I cannot see how one can obtain 1 min rainfall data (line 105) from a standard tipping-bucket rain

gauge. During many clock minutes, a bucket would simply be filling progressively, and might finally tip during a minute that actually delivered very little rain. This just requires some care in data processing. I generally don't see much validity in expressing tipping-bucket gauge data at any finer accumulation duration than 5 - 15 minutes, depending on the rainfall intensity.

**Response:** Sorry for the minor errors. Thank you so much for being so precise.

During the observation period, 226 natural rainfall events that generated runoff and soil erosion were recorded.

As shown in the following figure, in order to isolate hydrological disturbances from adjacent plots, concrete walls 100 cm deep were constructed around the runoff plots. Two runoff storage containers were set in the bottom of each plot to measure the surface and subsurface flow. The outlet of the subsurface flow was set at a soil depth of 60 cm at the bottom of the Bs layer.

Rainfall amount was measured with an accuracy of 0.2 mm using a tipping bucket rain gauge with a data logger. All rainfall data obtain methods have been modified in the revision text.

[Figure]

[Figure]

[Figure]

**Suggested revision in the text:**

The mistakes about natural rainfall events have corrected in lines 22, 321 and 504.
In lines 181-182, natural rainfall depth was measured by a tipping bucket rain gauge with a data logger (RG3-M, Onset Computer Corp., Bourne, MA, USA). The accuracy of rainfall data logging is recorded with the resolution of 0.2 mm (the rainfall amount under one tip).

In line 65, perhaps find a better expression than 'inter-fluctuations'. In line 166, avoid using '5-cm'. The international SI metric system requires that only a space can come between a quantity (in this case, 5) and the units of measurement (in this case, cm). The correct form is therefore "5 cm". In lines 171-172, I did not understand what the 5 measurements were for. Was this to measure the depth of water in a collecting vessel? In line 218 I think that the word 'and' is superfluous in the expression "the and runoff plot".

**Response:** We agree to correct it in the text. Thank you so much for being so precise.

**Suggested revision in the text:**

In line 65, we have replaced "inter-fluctuations in rainfall characteristics" with "rainfall intensity fluctuation".

In line 166, "5-cm" has been changed to "5 cm".

In lines 171-172, a ruler was used to measure the water levels repeating 5 times in each runoff container after each rainfall event as a means of calculating runoff volume.

In line 218, "the and runoff plot" has been changed to "the runoff plot".

Among the 'key words', I would suggest adding something like 'intensity profile' or 'storm type' or 'storm pattern' (the terminology is currently not settled).

**Response:** We agree to add the key words. Thank you so much for being so precise.

**Suggested revision in the text:** "Rainfall intensity fluctuation" and "natural rainfall " have been replaced with "rainfall intensity profile" and "storm pattern", respectively.

As the foregoing will have suggested, there is a considerable body of literature that could have been cited (e.g. on the time-sequence of rain events and the effect this has on antecedent wetness and hence on plot behavior). The authors might find the following paper to be of interest, for example, as it addresses the significance of intensity profile, though I do not suggest that this needs to be cited, as I am among the authors. Liang, Y., Gao, G., Liu, J., Dunkerley, D., & Fu, B. (2023). Runoff and soil loss responses of restoration vegetation under natural rainfall patterns in the Loess Plateau of China: The role of rainfall intensity fluctuation. CATENA, 225, 107013. doi:https://doi.org/10.1016/j.catena.2023.107013

**Response:** We are thankful for providing the latest research about the effects of natural rainfall patterns on water erosion. As stated by the reviewers, a number of scholars have examined the impact of the time-sequence of rain events on runoff and soil loss. For example, Mohamadi & Kavian (2015), Wang et al. (2017), Liu et al. (2022) and Liang et al. (2023) explored that the roles of natural rainfall pattern on surface runoff and soil loss. The erosive rainfall events were grouped into different rainfall patterns based on the occurrence time of the most intense rainfall, including the early-peak, center-peak, late-peak, and uniform patterns. The above studies shed light on the crucial role of natural rainfall pattern in yielding surface runoff and soil loss, and highlighted that which pattern may be responsible for triggering serious soil erosion.

However, little attention is given to the response of subsurface flow generation to natural rainfall patterns. Subsurface flow is a key component of rainfall runoff, and its output is even higher than that of surface runoff in the rainfall regime with long duration and high depth. The formation of subsurface flow significantly altered soil moisture redistribution, soil hydrology and slope erosion processes. Therefore, in this study, long-term in situ field observations of surface-subsurface flow and soil loss characteristics under natural rainfall events were conducted in three surface cover plots (bare land, litter and grass cover). This is the major novelty and advancement of this study. The results contributed to a better understanding of soil hydrological processes and erosion mechanisms caused by natural rainfall.

**Suggested revision in the text:** The literature (Liang et al., 2023) have been cited in the revision manuscript.

**References**

Dunkerley, D., 2021. The importance of incorporating rain intensity profiles in rainfall simulation studies of infiltration, runoff production, soil erosion, and related landsurface processes. Journal of Hydrology, 603, 126834.

Fang, N.F., Shi, Z.H., Li, L., et al., 2012. The effects of rainfall regimes and land use changes on runoff and soil loss in a small mountainous watershed. Catena, 99, 1-8.

Feng, J., Wei, W., Pan, D., 2020. Effects of rainfall and terracing-vegetation combinations on water erosion in a loess hilly area, China. Journal of Environmental Management, 261, 110247.

Liang, Y., Gao, G., Liu, J., Dunkerley, D., Fu, B., 2023. Runoff and soil loss responses of restoration vegetation under natural rainfall patterns in the Loess Plateau of China: The role of rainfall intensity fluctuation. Catena, 225, 107013.

Liu, J., Liang, Y., Gao, G., Dunkerley, D., Fu, B., 2022. Quantifying the effects of rainfall intensity fluctuation on runoff and soil loss: From indicators to models. Journal of Hydrology, 607, 127494.

Liu, Y.J., Yang, J., Hu, J.M., et al., 2016. Characteristics of the surface-subsurface flow generation and sediment yield to the rainfall regime and land-cover by long-term in-situ observation in the red soil region, Southern China. Journal of Hydrology, 539, 457-467.

Mohamadi, M.A., Kavian, A., 2015. Effects of rainfall patterns on runoff and soil erosion in field plots. International Soil and Water Conservation Research, 3, 273-281.

Wang, W., Yin, S., Xie, Y., Liu, B., Liu, Y., 2016. Effects of four storm patterns on soil loss from five soils under natural rainfall. Catena, 141, 56-65.

Wei, W., Chen, L., Fu, B., et al., 2007. The effect of land uses and rainfall regimes on runoff and soil erosion in the semi-arid loess hilly area, China. Journal of Hydrology, 335, 247-258.

Yang, Y., Zhu, R., Ma, D., et al., 2022. Multiple surface runoff and soil loss responses by sandstone morphologies to land-use and precipitation regimes changes in the Loess Plateau, China. Catena, 217, 106477.

---

## Author Comment (AC2)

---------------------RESPONSES TO THE COMMENTS--------------------

We appreciate the comments and provide our responses below to each comment. You will find your comments in **black**, while our responses are given in **blue** and any citation how we suggest to revise the text in the revised manuscript in **red**.

**Reviewer #2:**

**General comments:**

Rainfall is the most important driving force for runoff generation and erosion. The authors in this manuscript present a comparison experiment for runoff and erosion to discuss the distribution of rainfall volume and intensity within each event in different land covers based on long term in situ observations. The results are important to policy maker for flood and erosion preventions through optimized engineering measures as to land covers. The reviewer has some main concerns about the manuscript.

**Response:** We are thankful for the clear and constructive comments. We took them all into account in detail and responded each as listed below:

I don't know how do they fix up the litter cover, and have the soil pores and other structural attributes been altered through the years? I found hydrologic functions in the litter and grass plots are very close. It is beyond my intuitive expectations that litter covered bare soils with only 5cm thick layer of litter cannot change soil structures and then runoff generation mechanism naturally. Hence, they'd better provide more soil attributes in the profiles in Section 2.

**Response:** Thank you for the suggestions. For surface runoff and soil loss reduction, the grass cover and litter cover had equivalent hydrological functions. This similarity could be because the two cover types provided an effective buffer zone to protect the topsoil from splash erosion and weakened the kinetic energy of the raindrops and surface runoff. However, the grass cover and litter cover had different influence on subsurface flow. Compared to bare land, our results showed that the increasing benefit of subsurface flow for litter cover ranged from 1.38 to 2.67 times those of grass cover. The reasons for this phenomenon were as following:
(1) In the litter cover plot, a 5-cm thick layer of litter was placed on the soil surface to reduce water erosion. The litter was supplied from cutting *Paspalum natatum* Flugge and decomposed naturally. The litter was replenished quarterly throughout the observation period. Besides covering on soil surface, the plant litter was buried by soil particles from water and wind erosion (Hewins et al., 2017). The incorporation of plant litter into the topsoil layer actively exerts important effect on soil properties including bulk density, water stable aggregate, water repellency, and organic matter content (Figure 1), then increased rainfall infiltration and subsurface flow (Jordán et al., 2010; Wang et al., 2020; Yang et al., 2021).

[Figure]

Figure 1. Grass litter was incorporated into top-soil layer and improved soil structure.

(2) The plant litter incorporation depth into soil was less than grass root-soil systems. Soil macropore porosity and connectivity by decomposed litter mixed in the soil was quite different from the grass root channel. As a result, the preferential flow flux and depth in the litter cover plot were smaller than those of the grass cover plot (Guo et al., 2019). Therefore, for the litter cover, a large amount of rainfall was stored in the topsoil layer. Because of loose top-compact bottom soil configuration, the topsoil was easily saturated and generated subsurface flow, but rainfall was more difficult to convert into deep soil moisture. For the grass cover, before topsoil saturation, rainfall was highly susceptible to rapid infiltration into the subsoil by preferential flow, leading to little subsurface flow and large water storage in the deeper layers.

**Suggested revision in the text:** We agree to provide more soil properties along soil profiles in Section 2.

**Table 1**

Soils physic-chemical properties of different layers for the three runoff plots.

| Soil properties | Soil depth (cm) | Bare land | Litter cover | Grass cover |
|---|---|---|---|---|
| Bulk density | 0-30 | 1.35 | 1.25 | 1.19 |
| (g cm$^{-3}$) | 30-60 | 1.27 | 1.27 | 1.16 |
| | 60-90 | — | — | — |
| Soil porosity (%) | 0-30 | 46.64 | 51.17 | 53.33 |
| | 30-60 | 44.43 | 48.58 | 50..21 |
| | 60-90 | — | — | — |
| Soil organic matter | 0-30 | 11.38 | 14.81 | 19.87 |
| (g kg$^{-1}$) | 30-60 | 5.57 | 6.84 | 8.63 |
| | 60-90 | 4.93 | 4.26 | 5.24 |
| Water stable | 0-30 | 81.76 | 91.45 | 97.45 |
| macroaggregate (%) | 30-60 | 60.45 | 75.11 | 90.54 |
| | 60-90 | 57.79 | 65.67 | 82.63 |

Moreover, the reviewer suggests more properties describing intra-events should be provided and the authors should also confirm the main erosional rain patterns in the local area, the AD type? Whether the effects of rainfall intensity fluctuation on runoff are controlled by an intensity threshold? How does it function compared with inter-event properties? In addition, as the experimental plots are in the humid hilly regions, why forest cover was not discussed in the manuscript? What is the main runoff generation mechanism and erosion condition in local forest areas which maybe the main land covers?

**Response:** Thank you for the suggestions.

In order to explore the peak intensity differences among the four patterns, I30 and accumulative rainfall percent for each 1/3 period were provided in Table 1. As shown in Table 1, the I30 of advanced patterns was more than the other patterns. The results were the main reason for the highest soil loss amount in advanced patterns.

In lines 24-26, for bare land, the advanced pattern with the shortest duration and the highest intensity was main rainfall type for surface runoff and soil loss; the contribution rates were 57.24% and 75.17% for surface runoff and soil loss, respectively.

Rainfall is the main driver of runoff generation and soil erosion. inter- and intra-event variation significantly alters rainfall infiltration, runoff generation and erosion processes (Dunkerley, 2021; An et al., 2022). The impacts of natural rainfall on water erosion have been extensively studied at an inter-event scale; however, very few studies have explored the intra-event influences and associated responses to different surface cover types. Obviously, the effects of rainfall intensity fluctuation (intra-event) on runoff are controlled by rainfall intensity threshold (inter-event). Yuan (2019) and An (2022) conducted a concurrent in-depth investigation of all rainfall partitioning components at inter- and intra-event scales for two typical xerophytic shrubs in the Loess Plateau of China. The results demonstrated that inter-event rainfall partitioning amount and percentage depended more on rainfall amount, and rainfall intensity and duration controlled intra-event rainfall-partitioning variables. However, very few studies have explored the inter- and intra-event surface-subsurface flow and soil loss characteristics. This might be worth considering in our future work.

The red soil of China covers an area of 13% of the Chinese mainland and is of great importance in agricultural production. For a long period, the red soil region has been facing severe soil and water loss due to abundant rainfall, hilly terrain and unsustainable farming practices (Shi et al., 2014; Fang et al, 2017). In recent decades, with the rapid growth of human population (40% of China's population) and heavy pressure on productive soil resources, increasing amounts of barren land with a secondary community on the slopes have been transformed into cropland and orchards (Figure 2). The agricultural lands are exposed to serious risks of soil and water loss due to the intense soil disturbances from large-scale mechanized excavation and the lack of surface vegetation cover (Duan et al., 2020; 2021). To reduce rapidly water erosion, as a typical soil and water conservation measure, mulching with litter or living plants is widely used around the world to increase surface coverage (Figure 3, Shi et al., 2012; Duan et al., 2022). Runoff generation behaviours

and their driving mechanisms of different surface covers are therefore critical in reducing soil erosion and improving the water management level of red soil region. For forest cover without human disturbance, the soil erosion has been negligible compared to the agricultural lands. Therefore, in order to focus on clarifying the mechanism of surface cover on runoff generation and soil loss, the forest cover was not discussed in the manuscript.

[Figure]

Figure 2. Severe soil and water loss from cropland and orchards expansion.

[Figure]

Figure 3. Grass cover and litter cover are widely used in soil and water conservation of agricultural lands.

**Suggested revision in the text:**

**Table 1**

Rainfall eigenvalues of four intra-event rainfall patterns (IRP). D, P, and $I_{30}$ refer to rainfall duration, depth and maximum rainfall intensity in 30 min.

| RIP | Sample size | Rainfall percent for each 1/3 period (%) | | | D (min) | | | P (mm) | | | $I_{30}$ (mm h$^{-1}$) | | |
|---|---|---|---|---|---|---|---|---|---|---|---|---|---|
| | | 0~1/3 | 1/3~2/3 | 2/3~1 | Min | Max | Mean | Min | Max | Mean | Min | Max | Mean |
| Advanced | 109 | 74 | 15 | 11 | 27 | 4319 | 732 | 4.6 | 130.9 | 23.2 | 1.0 | 45.3 | 8.8 |
| Intermediate | 48 | 22 | 64 | 12 | 22 | 2972 | 978 | 7.1 | 72.1 | 26.2 | 0.9 | 22.9 | 6.0 |
| Delayed | 57 | 11 | 10 | 79 | 100 | 6191 | 1409 | 8.5 | 129.3 | 30.8 | 1.0 | 51.2 | 6.1 |
| Uniform | 12 | 33 | 36 | 31 | 426 | 2460 | 1362 | 11.7 | 43.5 | 21.4 | 1.0 | 5.7 | 2.9 |

**Specific Comments:**

P means page, and L means lines.

P2L23: What is the meaning of advanced, intermediate, delayed, and uniform patterns?

**Response:** In this paper, according to the timing of the most intense period of rainfall, rainfall events with more than 40% of the rainfall amount concentrated in the first, second and last third periods (duration) were defined as advanced, intermediate, and delayed patterns, respectively. The remaining events without obvious peaks and rainfall distributing uniformly over the duration were regarded as uniform pattern.

**Suggested revision in the text:** In lines 22-23, according to the timing of the most intense period of rainfall, 262 rainfall events were classified into four patterns: advanced, intermediate, delayed, and uniform patterns.

P2L31: Compared to what that subsurface flow was increased from 3.55 to 5.92 times? Subsurface flow in the bare land?

**Response:** Sorry for the minor error. We suggest to revise this statement as follows:

**Suggested revision in the text:** In lines 29-31, for all rainfall patterns, surface cover significantly reduced surface runoff and soil erosion by 88.01 to 91.69% and by 97.80 to 97.95%, respectively, while subsurface flow was increased from 3.55 to 5.92 times compared to the bare land.

P2L34-36: 'surface cover weakened the influences of intra-event rainfall variation on surface-subsurface flow and soil loss. The results demonstrated that intra-event rainfall variation had important effects on surface-subsurface flow and soil loss…' WHY? You mean that the influences of intra-event rainfall variation are weak since runoff is impacted by vegetation-soils?

**Response:** For bare land, the highest surface runoff and soil loss were found in the advanced pattern, which were 1.29 to 2.42 times and 2.52 to 39.78 times higher than those in the other three patterns. The advanced pattern was the main type for surface flow and soil loss in the bare land, with contribution rates of 57.24% and 75.17%, respectively. All of the above results indicated that the events with more rainfall concentrated in the early stages were more favourable for surface flow and soil loss.

Vegetation effectively increases rainfall infiltration and reduces surface runoff and soil erosion. For each rainfall pattern, long-term continuous in situ observations showed that litter cover and grass cover significantly reduced surface runoff and soil erosion and increased subsurface flow compared to bare land. Compared to bare land, the contributions of the advanced pattern to surface flow and soil loss for the surface cover decreased by 9.97-11.69% and 15.68-19.31%, respectively. The statistical results showed that the variation coefficients of surface runoff and soil erosion for bare land were the highest (0.93 and 1.73), and those were more than those for litter cover and grass cover (0.61 and 1.10, 0.65 and 1.44) when the rainfall patterns changed. This phenomenon occurred because surface cover effectively reduced splash erosion from rainfall concentration, and led to surface soils without sealing formation, a higher soil infiltration rate and subsequently lower runoff and sediment loss. In addition, the effective buffer layer on the soil surface increased the surface roughness, and delayed the overland flow velocity, thereby reducing the scouring ability of surface runoff on soil. The improvement in the soil anti-erosion ability was another important reason for this phenomenon.

P5L84-85: subsurface flow is the main runoff generation mechanism for forest catchments in humid climates. Why do you say little attention is given in this field?

**Response:** In the red soil hilly region of Southern China, the soil on agricultural lands is stratified owing the influences of artificial tillage and natural action; the top tillage layer is extremely loose, porous, and highly permeable, while the bottom layer is relatively compact and weakly permeable. Hence, infiltrated water accumulates in the tillage layer to form subsurface flow owing to the lateral slope (Ma et al., 2022). In addition, in the red soi region, long-duration rainfall events account for a large proportion in this region. It is the main erosive rainfall type in this area, which causes heavy soil loss. Long-duration rainfall typically induces greater infiltration and a higher soil water content, ensures that subsurface flow fully develops (Liu et al., 2016; Ma et al., 2023). The existence of subsurface flow changes the output pattern of slope flow, reduces the effective weight and cohesion of soil particles, and thus increases soil erodibility (Wang et al., 2020). Complex surface–subsurface hydrological conditions lead to severe soil loss in this region, but does not completely aware of the water erosion mechanism. However, most of the existing research focuses on the response of surface flow and soil erosion to rainfall change, and little attention is given to the response of subsurface flow generation.

Due to long-duration rainfall, hilly terrain and soil configuration, the subsurface flow from the other land use types such as cropland, grassland, and orchard cannot be ignored other than forest cover in this region. The red soil of China is of great importance in agricultural production. Soil erosion in agricultural lands is one of the main challenges facing this region. Mulching with litter or living plants is widely used around the world to increase surface coverage and reduce rapidly water erosion. Runoff generation behaviours and their driving mechanisms of different surface covers are therefore critical in reducing soil erosion and improving the water management level. For forest cover without human disturbance, the soil erosion has been negligible compared to the agricultural lands. In order to focus on clarifying the mechanism of surface cover on runoff generation and soil loss, the forest cover was not discussed in the manuscript.

P6L106: According to what standards, all the rainfall events have been classified into four types? Does these rainfall attributes have important impacts on runoff?

**Response:** Thank you for the suggestions. Our previous results showed that rainfall depth had a significantly positive relationship with the surface runoff coefficient, the erosion amount and the subsurface flow rate in the study area (Liu et al., 2016; Duan et al., 2020). Therefore, according to the timing of the most intense period of rainfall, 262 rainfall events were classified into four patterns: advanced, intermediate, delayed, and uniform patterns.

**Suggested revision in the text:** In lines 106, according to the timing of the most intense period of rainfall, 262 rainfall events were classified into four patterns: advanced, intermediate, delayed, and uniform patterns.

P9L159-163: why do you plant grass rather than trees in the hilly areas under humid sub-tropical climate? I believe that soil structures can be quite different in grass or tree covers, which lead to distinct runoff generation mechanisms.

**Response:** Vegetation is an effective technique to prevent water erosion, Previous studies showed that grass achieved favorable erosion control benefits in a shorter time compared to trees (Zhu et al., 2021; Duan et al., 2022). Planting grass is widely used in soil and water conservation efforts. Exploring the runoff generation and their driving mechanisms contributed to control soil erosion and optimize the water management.

P9L171: Five measurements? What?

**Response:** Thank you so much for being so precise. We revised the text accordingly as follows.

**Suggested revision in the text:** In lines 171-172, a ruler was used to measure the water levels repeating 5 times in each runoff container after each rainfall event as a means of calculating runoff volume.

P10L179-180: Cause you study the effects of intra-rainfall properties on runoff and sediment flux, whether it is here correct to separate successive rainfall event into two events.

**Response:** Runoff and sediment observation at event scale were widely used in analyzing the relationship between natural rainfall characteristics and water erosion. Specially, when the interval between two consecutive rainy periods was>6 h, there were considered as two separate rainfall events (Wischmeier, 1959; Huff, 1967; Renard et al., 1997). In this paper, for the lag of subsurface flow, an erosive rainfall event was not considered independent unless the interval times over 12 h or more after the last rainfall event (Liu et al., 2016; Duan et al., 2017). As a result, surface-subsurface flow and soil loss of 226 erosive rainfall events were obtained from field observations. Next, according to the timing of the most intense period of rainfall, 226 rainfall events were classified into four intra-event patterns: advanced, intermediate, delayed, and uniform patterns.

P10L183-191: you should verify the effects of these intra-event characters on runoff generation in the introduction.

**Response:** Thank you for the suggestions. We agree to verify the effects of these intra-event characters on runoff generation in the introduction.

**Suggested revision in the text:** In lines 82-83, some indicators have been proposed to quantify intra-event characters, and their relationships with runoff and soil loss were investigated. Dunkerley (2015) proposed the intra-event rainfall intermittency within rainfall events according to a rainfall intensity threshold, and suggested that the rainfall events with large intermittency might eliminate the surface ponding and recover the infiltrability capacity, which facilitated the transformation of rainfall into infiltration, consequently decreasing the overland flow. Todisco (2014) defined the rainfall bursts (peaks) using a threshold rainfall rate and employed the characteristics of rainfall peaks to recognize the erosive events in cultivated fallow based on pluviograph records. Field monitoring also found that every "intensity burst" during a natural rainfall event would lead to the corresponding peak of surface runoff and sediment from coal

mining refuse site (Smith and Olyphant, 1994). Dunkerley (2020) proposed an index to describe rainfall amount in wettest 5% rainfall duration (EDf5), and the EDf5 might play a more important role in erosive processes compared to I30. Liu et al (2022) proposed that the average intensity of rainfall peaks (Iap) and rainfall duration of high-intensity zone (RDh) were the main variables to simulate runoff coefficient, and the average intensity of high-intensity zone (Iah) and relative amplitude of rainfall intensity (Ram) controlled sediment concentration and soil loss coefficient.

Dunkerley, D., 2015. Intra-event intermittency of rainfall: an analysis of the metrics of rain and no-rain periods. Hydrol. Process. 29 (15), 3294–3305.

Dunkerley, D., 2020. Rainfall intensity in geomorphology: challenges and opportunities. Prog. Phys. Geogr. 17, 1–26.

Liu, J., Liang, Y., Gao, G., Dunkerley, D., Fu, B., 2022. Quantifying the effects of rainfall intensity fluctuation on runoff and soil loss: From indicators to models. Journal of Hydrology 607, 127494.

Smith, L.C., Olyphant, G.A., 1994. Within-storm variations in runoff and sediment export from a rapidly eroding coal-refuse deposit. Earth Surf. Process. Landf. 19 (4), 369–375.

Todisco, F., 2014. The internal structure of erosive and non-erosive storm events for interpretation of erosive processes and rainfall simulation. J. Hydrol. 519, 3651–3663.

P10L198-199: how do you divide a rainfall event into three periods? It is not very clear to the readers.

**Response:** Erosive storms derived from pluviograph records were divided into four patterns using the following steps. Step 1: Instant rainfall and duration were divided by the total rainfall and duration, respectively, to make them dimensionless. Step 2: Dimensionless rainfall and duration were accumulated until they reached a value of one. Step 3: The dimensionless durations were separated into three equal time periods and accumulated rainfall for each period was calculated. Step 4: Storm patterns were defined by the location of maximum rainfall accumulation: (1) the advanced pattern was defined as more than 40% of the rainfall concentrating during the first third period, (2) the intermediate pattern was defined as more than 40% of the rainfall concentrating during the second third period, (3) the delayed pattern was defined as more than 40% of the rainfall concentrating during the last third period, and (4) the uniform pattern was defined as rainfall being uniformly distributed over the duration with no obvious peaks (Fig. 3). For example, one storm with 35, 32 and 33% of total rainfall amount in each third period was relatively uniform and was identified as the uniform pattern. The '40%' criterion was an upper boundary of largest rainfall period for the uniform pattern and a lower boundary for the other three patterns.

**Suggested revision in the text:** In lines 195-196, secondly, the dimensionless rainfall duration was divided into three equal parts, and the cumulative rainfall was calculated for each equal time period.

P13L238-239: how do you obtain the results?

**Response:** As shown in Table 1 of manuscript, the average rainfall depths of advanced, intermediate, delayed and uniform patterns were 23.2, 26.2, 30.8, and 21.4 mm, respectively, Therefore, the average rainfall amounts for different intra-event rainfall patterns were ranked in the order of delayed > intermediate > advanced > uniform.

P13L240-243: what is the meaning of SD in table 1?

**Response:** Sorry. In table 1, SD refers to the standard deviation.

P15L278: Why SSL in the grass cover plot (1.98 L mm -1) is smaller than the litter cover plot (2.98 L mm-1)?

**Response:** Different measures covering the soil surface had varying near-surface characteristics and effects on soil properties along the profile, resulting in disinct soil hydrological responses. The increasing benefit of subsurface flow for litter cover for each rainfall pattern was 1.38 to 2.67 times greater than that of grass cover. Due to no difference in surface runoff, more rainfall was converted to soil water storage and deeper infiltration under grass cover than under litter cover for constant rainfall. For long-term coverage of the soil surface, the plant litter was buried by soil particles from water and wind erosion. The incorporation of plant litter into the topsoil layer actively impacted soil hydraulic properties such as bulk density, soil infiltration, and saturated hydraulic conductivity. Rainfall infiltration tended to take the main form of matrix infiltration, with large amounts of rainfall being stored in the topsoil layer. Because of loose top- compact bottom soil configuration, the topsoil was easily saturated and generated more subsurface flow, but rainfall was more difficult to convert into deep soil moisture. For grass cover, rainfall infiltration was greater in main form of preferential flow with many root channels. Before topsoil saturation, rainfall was highly susceptible to rapid infiltration into the subsoil by preferential flow, leading to little subsurface flow and large water storage in the deeper layers. The related details were discussed in discussion (lines 438-453).

P17L318-320: Except for the rainfall intensity fluctuation, whether other properties of intra-event will also impact runoff generation? How does it function compared with inter-event properties?

**Response:** The temporal sequence of fluctuating rainfall intensities during an event - the intensity profile or 'rainfall pattern' - is known to affect water partitioning at the ground surface and associated erosional processes. There can be large differences in runoff ratios and runoff rates between early-peak and late-peak events, and hence in the depth, speed, and sediment transport capacity of resulting overland flow. Classification based on rainfall profiles is important for describing intra-event rainfall variation. Except for rainfall intensity fluctuation, rainfall continuity or intermittency also, as breaks in rainfall can be critical in allowing overland flow to slow and for soil to be re-deposited, for ephemeral surface ponding to dissipate, for soil infiltrability to partially recover, and so on. Periods of reduced rainfall intensity (or of true intermittency) allow

soil infiltrability to partially recover toward pre-rain values (Dunkerley 2018), permit surface ponding to partially or wholly dissipate (Aryal et al., 2007) and allow some drying of vegetation canopies (Kume et al., 2008), understory vegetation and ground litter, so regenerating rainfall interception capacity. We suggest to discuss the effects of rainfall intermittency on runoff and sediment for the four rainfall patterns.

**Suggested revision in the text:** The characteristic intermittency of rainfall includes temporary cessations (hiatuses), as well as periods of very low intensity within more intense events (Dunkerley, 2018; 2021). For instance, Figure 3 clearly shows that there is ubiquitous intermittency in most natural rainfall events, especially for long-duration rainfall events. The advanced pattern events might have been late afternoon convective events. Intense rainfall intensity was concentrated in the early period, while intermittency often occurs in the later periods (Figure 3). The intense intensity in the advanced pattern tended to induce strong runoff scour in the early periods, which greatly contributed to the development of concentrated flow and produced more intense soil erosion. In addition, early intense rainfall may also induce more splash erosion, which also provides abundant loose soil particles and thus leads to greater sediment production capacity. Conversely, the delayed pattern might have been overnight falls with long time. Most of rainfall interval occurred in the early and middle stages (Figure 3), and therefore the apparent infiltration rate expands, which greatly increased the time to first runoff and reduced surface runoff and soil erosion in the early stages. In addition, the subsurface flow was increased due to the increase of infiltration rate and intense rainfall intensity in later periods.

(1) Dunkerley, D., 2018. How is overland flow produced under intermittent rain? An analysis using plot-scale rainfall simulation on dryland soils. Journal of Hydrology, 556, 119–130.

(2) Dunkerley, D., 2021. Intermittency of rainfall at sub-daily timescales: New quantitative indices based on the number, duration, and sequencing of interruptions to rainfall. Atmospheric Research, 253, 105475.

P17L343-344: 'Dunkerley (2012) determined that uniform events of unvarying intensity yielded the lowest total runoff.' The selected events of Dunkerley are with uniform but low rainfall intensity, aren't he?

**Response:** Dunkerley (2012) designed primarily to understand how intensity profile affects infiltration and runoff, all events, regardless of intensity profile, had the same depth, duration, and average intensity. In that work, all events lasted 90 minutes and delivered 15 mm of rainfall at an average rainfall rate of 10 mm h$^{-1}$. This ability to hold depth and duration constant experimentally isolates, at least to some extent, the intensity profile itself as the factor than could account for differences in plot infiltration and runoff. The results demonstrated that event profile (peak intensity in which period) exerted an important effect on infiltration and runoff. 'Uniform' events of unvarying intensity yielded the lowest total runoff, the lowest peak runoff rate and the lowest runoff ratio (0.13). Compared with 'uniform' runs, the varying intensity runs yielded larger runoff ratios and peak runoff rates, exceeding those of the 'uniform' events by 85%–570%.

P19L358-366: the reviewer suggests that the authors should provide and discuss rainfall intensity and volume for each pattern.

**Response:** Thank you for the suggestions. We agree to discuss the effects of rainfall intensity and volume for each pattern on surface-subsurface flow and soil loss.

**Suggested revision in the text:** In lines 358-366, the correlations of surface runoff coefficient (ROC), subsurface flow rate (SSL), and soil loss rate (SLR) with rainfall parameters were provided in the discussion.

**Table 2**

Statistical analysis of the surface runoff coefficient (ROC) with rainfall parameters.

| | surface cover | depth | | duration | | I | | I30 | |
|---|---|---|---|---|---|---|---|---|---|
| | | r | p | r | p | r | p | r | p |
| advanced | bare land | **0.471\*\*** | 0.000 | -0.082 | 0.395 | **0.360\*\*** | 0.000 | **0.582\*\*** | 0.000 |
| | litter cover | **0.496\*\*** | 0.000 | 0.156 | 0.104 | 0.143 | 0.139 | **0.265\*\*** | 0.005 |
| | grass cover | **0.488\*\*** | 0.000 | 0.141 | 0.142 | 0.176 | 0.066 | **0.328\*\*** | 0.000 |
| intermediate | bare land | **0.356\*** | 0.013 | 0.201 | 0.170 | 0.156 | 0.291 | **0.387\*\*** | 0.007 |
| | litter cover | **0.499\*\*** | 0.000 | 0.108 | 0.466 | 0.066 | 0.658 | **0.428\*\*** | 0.002 |
| | grass cover | **0.522\*\*** | 0.000 | 0.164 | 0.267 | 0.134 | 0.365 | **0.439\*\*** | 0.002 |
| delayed | bare land | **0.358\*\*** | 0.006 | -0.063 | 0.640 | 0.213 | 0.111 | **0.289\*** | 0.029 |
| | litter cover | **0.276\*** | 0.037 | -0.001 | 0.992 | 0.224 | 0.094 | -0.009 | 0.949 |
| | grass cover | **0.447\*\*** | 0.000 | 0.036 | 0.789 | **0.289\*** | 0.030 | 0.115 | 0.393 |
| uniform | bare land | 0.382 | 0.220 | 0.156 | 0.628 | 0.102 | 0.752 | 0.563 | 0.057 |
| | litter cover | 0.508 | 0.092 | -0.142 | 0.660 | 0.504 | 0.095 | **0.653\*** | 0.021 |
| | grass cover | 0.573 | 0.051 | -0.048 | 0.882 | 0.367 | 0.240 | 0.343 | 0.274 |

**Table 3**

Statistical analysis of the subsurface flow rate (SSL) with rainfall parameters.

| | surface cover | depth | | duration | | I | | I30 | |
|---|---|---|---|---|---|---|---|---|---|
| | | r | p | r | p | r | p | r | p |
| advanced | bare land | **0.286\*\*** | 0.003 | **0.374\*\*** | 0.000 | -0.101 | 0.295 | -0.094 | 0.332 |
| | litter cover | **0.656\*\*** | 0.000 | **0.340\*\*** | 0.000 | -0.038 | 0.697 | 0.156 | 0.106 |
| | grass cover | **0.459\*\*** | 0.000 | **0.261\*\*** | 0.006 | -0.020 | 0.835 | 0.133 | 0.166 |
| intermediate | bare land | **0.470\*\*** | 0.001 | **0.418\*\*** | 0.003 | -0.131 | 0.375 | 0.067 | 0.651 |
| | litter cover | **0.327\*** | 0.023 | 0.238 | 0.104 | -0.107 | 0.469 | 0.053 | 0.722 |
| | grass cover | **0.360\*** | 0.012 | 0.266 | 0.068 | -0.086 | 0.563 | 0.118 | 0.426 |
| delayed | bare land | 0.229 | 0.087 | 0.071 | 0.600 | 0.209 | 0.119 | -0.025 | 0.854 |
| | litter cover | 0.243 | 0.069 | 0.056 | 0.679 | 0.100 | 0.459 | 0.090 | 0.505 |
| | grass cover | 0.124 | 0.358 | 0.075 | 0.579 | -0.059 | 0.662 | -0.012 | 0.929 |
| uniform | bare land | **0.844\*\*** | 0.0006 | 0.189 | 0.556 | 0.317 | 0.315 | **0.661\*** | 0.019 |
| | litter cover | -0.019 | 0.954 | 0.143 | 0.658 | -0.190 | 0.555 | 0.149 | 0.645 |
| | grass cover | 0.396 | 0.203 | 0.345 | 0.272 | -0.100 | 0.756 | 0.463 | 0.130 |

**Table 4**

Statistical analysis of the soil loss rate (SLR) with rainfall parameters.

| | surface cover | depth | | duration | | I | | I30 | |
|---|---|---|---|---|---|---|---|---|---|
| | | r | p | r | p | r | p | r | p |
| advanced | bare land | **0.598**\** | 0.000 | -0.019 | 0.847 | **0.351**\** | 0.000 | **0.718**\** | 0.000 |
| | litter cover | **0.512**\** | 0.000 | 0.017 | 0.859 | **0.380**\** | 0.000 | **0.525**\** | 0.000 |
| | grass cover | **0.212**\* | 0.027 | 0.000 | 0.997 | **0.320**\** | 0.0007 | **0.285**\** | 0.003 |
| intermediate | bare land | **0.565**\** | 0.000 | 0.162 | 0.271 | 0.218 | 0.136 | **0.802**\** | 0.000 |
| | litter cover | 0.218 | 0.136 | 0.041 | 0.784 | 0.281 | 0.053 | 0.281 | 0.053 |
| | grass cover | -0.080 | 0.590 | -0.189 | 0.198 | **0.344**\* | 0.017 | 0.164 | 0.267 |
| delayed | bare land | **0.629**\** | 0.000 | 0.105 | 0.436 | 0.180 | 0.181 | **0.781**\** | 0.000 |
| | litter cover | 0.221 | 0.099 | -0.077 | 0.568 | 0.005 | 0.972 | **0.376**\** | 0.004 |
| | grass cover | **0.368**\** | 0.005 | 0.103 | 0.449 | 0.040 | 0.772 | **0.321**\* | 0.016 |
| uniform | bare land | **0.632**\* | 0.028 | 0.280 | 0.378 | 0.135 | 0.677 | **0.690**\* | 0.013 |
| | litter cover | 0.049 | 0.879 | 0.131 | 0.684 | -0.007 | 0.983 | -0.202 | 0.530 |
| | grass cover | 0.516 | 0.086 | 0.184 | 0.567 | 0.207 | 0.520 | 0.316 | 0.317 |

P22L395-396: The delayed and uniform patterns were characterized by long-duration and heavy-intensity rainfall. In fact, they are not so heavy and long. In L381, the authors claimed that the uniform events with long duration and low intensity……It is in fact very complex even for rainfall events in the same pattern. So, the question is how to compared rainfall-runoff event objectively with unified standards.

**Response:** In order to eliminate the influence of rainfall depth variance, three variables, surface runoff coefficient (ROC), subsurface flow rate (SSL), and soil loss rate (SLR) were employed to characterize the effect of intra-event rainfall patterns on water erosion. The ROC was calculated as the ratio of runoff depth to rainfall amount (%). The SSL referred to the subsurface flow rate of an erosive rainfall event (L mm$^{-1}$), which was calculated by the ratio of subsurface flow volume and rainfall amount. The SLR was defined as the sediment yield in unit area induced by unit rainfall amount (t km$^{-2}$ mm$^{-1}$), which could substantially demonstrate the effects of rainfall pattern excluding the effects of rainfall amount. The ROC, SSL and SLR reflected the ability of rainfall to produce surface-subsurface flow and soil loss, respectively. Therefore, the rainfall-runoff events were compared objectively with unified standards.

**Suggested revision in the text:** In line 395-396, table 1 clearly shows that the delayed and uniform patterns were characterized by long-duration.

P22L406-407: which pattern (AD or UF) is more like to increase risk for erosions as the authors claimed AD events tend to increase erosional risks in above.

**Response:** We agree to revise the sentence.

**Suggested revision in the text:** In lines 406-407, as a result, in terms of soil erosion risk, in addition to advanced rainfall with a short duration and high intensity, more attention should be paid to delayed and uniform rainfall events with long duration.

P24L438-439: the reviewer suggest bedrock depth or soil thickness and soil pores and hydraulic conductivity should be provided here.

**Response:** We agree to provide the soil thickness and soil hydraulic properties. The soil thickness of the study area was provided in lines 132-143. Specially, the dominant soil type of the region is red clay soil, which is formed by the decomposition of Quaternary sediments. Red clay soil is classified as Ultisol in the USDA soil taxonomy system, and it is vulnerable to water erosion. This soil has a texture composed of $11.54\pm1.21\%$ sand (2-0.05 mm), $68.06\pm0.15\%$ silt (0.05-0.002 mm), and $20.41\pm1.19\%$ clay (<0.002 mm). The soil thickness typically exceeds 100 cm, and the soil profile type is Ah-Bs-Cs according to Soil Taxonomy (Liu et al., 2016; Ma et al., 2022). The soil physicochemical properties vary considerably among the different layers, especially regarding soil porosity and water infiltration capacity. The topsoil layer (Ah) is typically 30 cm, and it is susceptible to severe soil erosion because of its loose structure ($1.27\pm0.10$ g cm$^{-3}$) and the high precipitation in this region. The depth of the Bs layer is 30-60 cm with a compact structure ($1.42\pm0.08$ g cm$^{-3}$) and low permeability. The soil below 60 cm is defined as the Cs layer (parent material) with a tight structure ($1.53\pm0.07$ g cm$^{-3}$) and poor permeability.

**Suggested revision in the text:** We agree to provide more soil properties along soil profiles.

**Table 5**

Topsoil physic-chemical properties for the three runoff plots.

| Soil properties | Bare land | Litter cover | Grass cover |
|---|---|---|---|
| Bulk density (g cm$^{-3}$) | 1.35 | 1.25 | 1.19 |
| Total porosity (%) | 46.64 | 51.17 | 53.33 |
| Capillary porosity (%) | 41.39 | 40.11 | 41.70 |
| Field capacity (%) | 19.14 | 24.65 | 27.42 |
| Saturated water content (%) | 34.55 | 40.94 | 44.82 |
| Soil organic matter (g kg$^{-1}$) | 11.38 | 14.81 | 19.87 |
| Water stable macroaggregate (%) | 81.76 | 91.45 | 97.45 |

P25L454: the reviewer argue that soil structures can be quite different between Fig.10a & b. so the figure shall be corrected.

**Response:** We agree to correct the Fig.10a & b. Thank you so much for being so precise.

**Suggested revision in the text:** Figure 10

[Figure]

**Figure 10.** Conceptual diagram of the soil hydrological processes under the bare land (a), litter cover (b) and grass cover (c) slopes.

**References**

Aryal, R.K., Furumai, H., Nakajima, F., Jinadasa, H., 2007. The role of inter-event time definition and recovery of initial/depression loss for the accuracy in quantitative simulations of highway runoff. Urban Water J. 4 (1), 53 –58.

Duan, J., Liu, Y.J., Yang, J., Tang, C.J., Shi, Z.H., 2020. Role of groundcover management in controlling soil erosion under extreme rainfall in citrus orchards of southern China. J. Hydrol. 582, 124290.

Duan, J., Liu, Y.J., Tang, C.J., Shi, Z.-H., Yang, J., 2021. Efficacy of orchard terrace measures to minimize water erosion caused by extreme rainfall in the hilly region of China: Long-term continuous in situ observations. J. Environ. Manage. 278, 111537.

Duan, J., Liu, Y.J., Wang, L.Y., Yang, J., Tang, C.J., Zheng, H.J., 2022. Importance of grass stolons in mitigating runoff and sediment yield under simulated rainstorms. Catena, 213, 106132.

Duan, J., Yang, J., Tang, C.J., Chen, L.H., Liu, Y.J., Wang, L.Y., 2017. Effects of rainfall patterns and land cover on the subsurface flow generation of sloping Ferralsols in southern China. PLoS One 12, e0182706.

Dunkerley, D., 2012. Effects of rainfall intensity fluctuations on infiltration and runoff: rainfall simulation on dryland soils, Fowlers Gap, Australia. Hydrol. Process. 26, 2211–2224.

Dunkerley, D., 2018. How is overland flow produced under intermittent rain? An analysis using plot-scale rainfall simulation on dryland soils. J. Hydrol. 556, 119–130.

Fang, N.F., Wang, L., Shi, Z.H., 2017. Runoff and soil erosion of field plots in a subtropical mountainous region of China. Journal of Hydrology, 552, 387–395.

Hewins, D.B., Sinsabaugh, R.L., Archer, S.R., Throop, H.L., 2017. Soil litter mixing and microbial activity mediate decomposition and soil aggregate formation in a sandy shrub-invaded Chihuahuan Desert grassland. Plant Ecol. 218(4), 459–474.

Huff, F.A., 1967. Time distribution of rainfall in heavy storms. Water Resour. Res. 3 (4), 1007–1019.

Jordán, A., Zavala, L.M., Gil, J., 2010. Effects of mulching on soil physical properties and runoff under semi-arid conditions in southern Spain. Catena 81, 77–85.

Kume, T., Manfroi, O.J., Suzuki, M., Tanaka, K., Kuraji, K., Nakagawa, M., Kumagai, T.o., 2008. Estimation of vertical profiles of leaf drying times after daytime rainfall within a Bornean tropical rainforest. Hydrol. Process. 22 (18), 3689–3696.

Liu, Y.J., Yang, J., Hu, J.M., Tang, C.J., Zheng, H.J., 2016. Characteristics of the surface-subsurface flow generation and sediment yield to the rainfall regime and land-cover by long-term in-situ observation in the red soil region, Southern China. Journal of Hydrology 539, 457–467.

Ma, Y., Li, Z., Deng, C., Yang, J., Tang, C., Duan, J., Zhang, Z., Liu, Y., 2022. Effects of tillage-induced soil surface roughness on the generation of surface–subsurface flow and soil loss in the red soil sloping farmland of southern China. Catena, 213, 106230.

Ma, Y., Liu, Y., Tian, L., Long, Y., Lei, M., Duan, J., Yang, J., Nie, X., Li, Z., 2023. Roles of soil surface roughness in surface–subsurface flow regulation and sediment sorting. Journal of Hydrology, 623, 129834.

Prosdocimi, M., Cerdà, A., Tarolli, P., 2016. Soil water erosion on Mediterranean vineyards: a review. Catena, 141, 1–21.

Renard, K.G., Foster, G.R., Weesies, G.A., McCool, D.K., Yoder, D.C., 1997. Predicting Soil Erosion by Water. U.S. Department of Agriculture, Agricultural Research Service, Agriculture Handbook 703.

Shi, Z.H., Yue, B.J., Wang, L., Fang, N.F., Wang, D., Wu, F.Z., 2012. Effects of much cover rate on interrill erosion processes and the size selectivity of eroded sediment on steep slopes. Soil Sci. Soc. Am. J. 77, 257–267.

Shi, Z.H., Huang, X.D., Ai, L., Fang, N.F., Wu, G.L., 2014. Quantitative analysis of factors controlling sediment yield in mountainous watersheds. Geomorphology 226, 193–201.

Wang, C.F., Wang, B., Wang, Y.J., Wang, Y.Q., Zhang, W.L., Zhang, X.J., 2020. Rare earth elements tracing interrill erosion processes as affected by near-surface hydraulic gradients. Soil and Tillage Research, 202, 104673

Wang, L.J., Zhang, G.H., Zhu, P.Z., Wang, X., 2020. Comparison of the effects of litter covering and incorporation on infiltration and soil erosion under simulated rainfall. Hydrol. Process. 34, 2911–2922.

Wischmeier, W.H., Smith, D.D., 1965. Predicting Rainfall-erosion Losses From Cropland East of the Rocky Mountains. U.S. Department of Agriculture, Agricultural Research Service, Agriculture Handbook 282.

Yang, J., Liu, H., Lei, T., Rahma, A.E., Liu, C., Zhang, J., 2021. Effect of straw-incorporation into farming soil layer on surface runoff under simulated rainfall. Catena, 199, 105082.

Zhu, P.Z., Zhang, G.H., Wang, H.X., Yang, H.Y., Zhang, B.J., Wang, L.L., 2021. Effectiveness of typical plant communities in controlling runoff and soil erosion on steep gully slopes on the Loess Plateau of China. Journal of Hydrology, 602, 126714.